# The ion selectivity filter is not an activation gate in TRPV1-3 channels

Andrés Jara-Oseguera*, Katherine E Huffer, Kenton J Swartz*

Molecular Physiology and Biophysics Section, Porter Neuroscience Research Center, National Institute of Neurological Disorders and Stroke, National Institutes of Health, Bethesda, United States

**Abstract** Activation of TRPV1 channels in sensory neurons results in opening of a cation permeation pathway that triggers the sensation of pain. Opening of TRPV1 has been proposed to involve two gates that appear to prevent ion permeation in the absence of activators: the ion selectivity filter on the external side of the pore and the S6 helices that line the cytosolic half of the pore. Here we measured the access of thiol-reactive ions across the selectivity filters in rodent TRPV1-3 channels. Although our results are consistent with structural evidence that the selectivity filters in these channels are dynamic, they demonstrate that cations can permeate the ion selectivity filters even when channels are closed. Our results suggest that the selectivity filters in TRPV1-3 channels do not function as activation gates but might contribute to coupling structural rearrangements in the external pore to those in the cytosolic S6 gate.

## Introduction

Transient Receptor Potential (TRP) ion channels are involved in multiple important biological processes in organisms from yeast to mammals (*Li, 2017*). These processes range from organelle function and trafficking in single cells (*Zhang et al., 2018b*) to the primary detection of sensory stimuli (*Mickle et al., 2015*) and whole-animal thermoregulation (*Wang and Siemens, 2015*). TRPV1 is particularly important due to its role as an integrator of pain-producing stimuli and inflammatory mediators in nociceptive sensory neurons (*Moore et al., 2018*). Underlying all these processes is the ability of TRP channels to rapidly conduct cations across biological membranes in response to remarkably diverse types of activating stimuli, either for net cation transport or to trigger electrical signaling.

Vertebrates express 27 TRP channel genes, grouped into six subfamilies: TRPA – Ankyrin; TRPC – Canonical; TRPM – Melastatin; TRPML – Mucolipin; TRPP – Polycystin; TRPV – Vanilloid. TRP channels assemble as homo- or hetero-tetramers with cytosolic domains that differ in size and structure between subfamilies, and a transmembrane domain fold that is very similar between all members of the family. Each subunit contains six transmembrane helices (S1-S6) that fold in a similar way to voltage-gated cation channels (*Madej and Ziegler, 2018*). The S1-S4 helices from each subunit form modulatory domains that surround the central cation-conducting pathway formed by four sets of S5 and S6 helices (*Figure 1*) (*Zhang et al., 2018b*).

Stimuli that activate TRP channels target sites throughout the entire protein (*Latorre et al., 2009*), yet all must ultimately alter the structure of the pore to enable it to open. Structural data suggest that the general mechanism for opening and closing (i.e. gating) the pore could be conserved throughout the TRP channel family (*Palovcak et al., 2015*; *Kasimova et al., 2018*). In particular, the pore domain in most TRP channel structures adopts a non-conducting conformation in which the cytosolic half of the ion conduction pathway is obstructed by the pore-lining S6 helices at one or more relatively conserved cytosolic constrictions that could function as an activation gate (*Figure 1* and *Figure 1—figure supplements 1* and *2*). As expected for an activation gate, this cytosolic constriction expands in some TRP channel structures that were solved in the presence of activators or

*For correspondence:
andres.jara-oseguera@nih.gov (AJO);
swartzk@ninds.nih.gov (KJS)

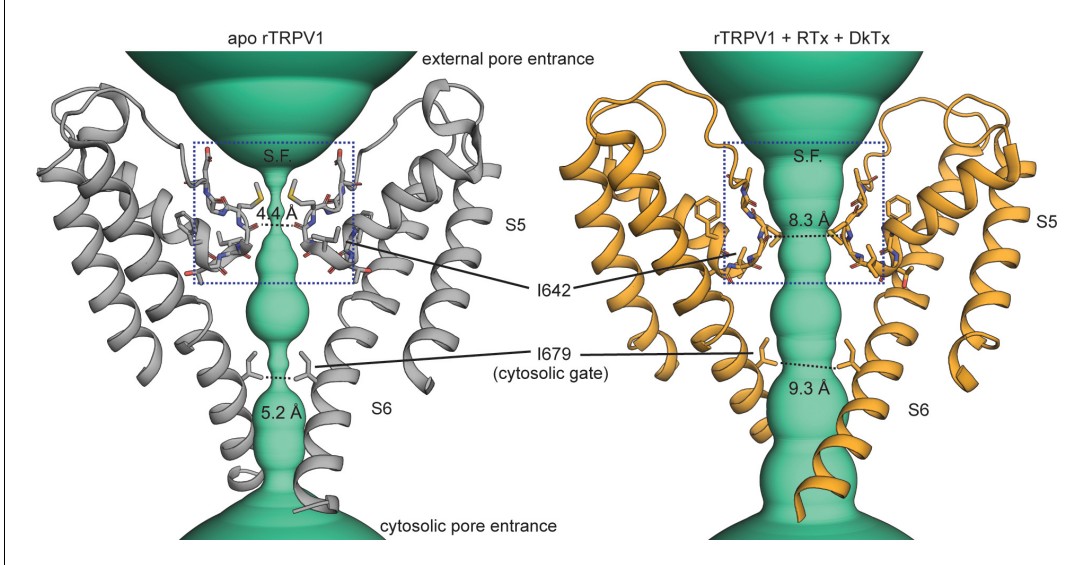

**Figure 1.** Gating the pore of the TRPV1 channel. The pore domain of the unliganded (apo, PDB: 5IRZ) and Resiniferatoxin (RTx)/Double-knot toxin (DkTx) bound (PDB: 5IRX) TRPV1 channel structures (*Gao et al., 2016b*) with only two opposing subunits depicted. The turquoise densities illustrate the ion-conducting pathway according to the van der Waals pore-radius calculated using HOLE (*Smart et al., 1996*), which is explicitly shown at the narrowest points on the selectivity filter (S.F.) and cytosolic gate.

The online version of this article includes the following figure supplement(s) for figure 1:

**Figure supplement 1.** The ion-conducting pathway of the TRPV2 and TRPV3 channels.

**Figure supplement 2.** Constrictions in the ion-conduction pathway of TRP channel structures.

mutations that favor opening (*Cao et al., 2013*; *Gao et al., 2016b*; *Schmiege et al., 2017*; *Wilkes et al., 2017*; *Huang et al., 2018*; *Hughes et al., 2018*; *McGoldrick et al., 2018*; *Singh et al., 2018*; *Wang et al., 2018*; *Zheng et al., 2018b*; *Zubcevic et al., 2019a*) (*Figure 1* and *Figure 1—figure supplements 1* and *2*). Moreover, functional assays have confirmed the presence of a cytosolic gate in the TRPV1 channel (*Oseguera et al., 2007*; *Jara-Oseguera et al., 2008*; *Salazar et al., 2009*), as well as other TRPV, TRPC, TRPM and TRPP channels (*Zheng et al., 2018a*; *Zheng et al., 2018b*).

Structures of TRPV1 (*Gao et al., 2016b*), TRPV2 (*Zubcevic et al., 2018b*; *Dosey et al., 2019*; *Zubcevic et al., 2019b*) and TRPP1 (*Shen et al., 2016*) channels in different states have additionally suggested that the selectivity filter, which 'selects' the types of cations that can permeate upon channel activation (*Owsianik et al., 2006*), can physically obstruct cation permeation (*Figure 1* and *Figure 1—figure supplement 1A*). TRPV1 channels are activated by multiple stimuli that target the external face of the pore where the selectivity filter resides (*Jordt et al., 2000*; *Bohlen et al., 2010*; *Bae et al., 2016*; *Gao et al., 2016b*; *Jara-Oseguera et al., 2016*). In addition, mutations in that region severely impact gating in TRPV (*Grandl et al., 2008*; *Myers et al., 2008*; *Grandl et al., 2010*; *Zhang et al., 2019*), TRPA1 (*Wang et al., 2013*; *Kurganov et al., 2017*) and TRPM (*Oberwinkler et al., 2005*; *Tóth and Csanády, 2012*; *Zhang et al., 2018c*) channels. Together, these results suggest that the selectivity filter region has an important influence on channel activation and that the filter could act as an activation gate in addition to that formed by the S6 helices. Indeed, the selectivity filter is directly involved in the gating of channels that have a similar transmembrane fold to TRP channels: Gating at the filter controls stimulus-dependent opening and closing of large-conductance calcium-activated K$^+$ (BK) channels (*Wilkens and Aldrich, 2006*; *Zhou et al., 2011*), cyclic-nucleotide activated channels (CNG) (*Contreras and Holmgren, 2006*; *Contreras et al., 2008*) and two-pore domain K$^+$ (K2P) channels (*Bagriantsev et al., 2011*; *Piechotta et al., 2011*). Alternatively, the filter could restrict permeation through TRP channels during inactivation (*Tóth and Csanády, 2012*), as is the case in voltage-gated K$^+$ (*Smith et al., 1996*; *Hoshi and Armstrong, 2013*), Ca$^{2+}$ (*Abderemane-Ali et al., 2019*) and possibly Na$^+$ (*Ong et al., 2000*; *Xiong et al., 2006*; *Catterall et al., 2017*) channels. In fact, some K$^+$ channels can be robustly

opened by drugs that target the selectivity filter, regardless of whether it controls activation or inactivation (*Schewe et al., 2019*).

Here we investigate the accessibility of pore-lining residues in TRPV1 channels to determine whether the selectivity filter is indeed an activation gate. We establish that the filter adopts ion-conducting conformations in the absence of agonists, revealing that it does not act as a gate and underscoring the importance of the S6 helices for channel gating. We obtain similar results with TRPV2 and TRPV3 channels, which share activation properties with TRPV1 but have distinct pharmacology (*Yang et al., 2016*; *Zhang et al., 2016*; *Zhang et al., 2019*). Because of the known importance of the selectivity filter region for channel activation, and the conformational changes observed in that region for structures of TRPV channels solved under different conditions, we propose that, rather than acting as a gate, the filter functions as an actuator to couple structural rearrangements in the external pore to those in the cytosolic S6 gate.

## Results

### Ag⁺ blocks TRPV1 channels in a state-dependent manner

If the selectivity filter of TRPV1 functions as an activation gate, pore-lining residues below the filter will only be accessible to external cations (Na$^+$ in our recording conditions) when the channel is open. We chose to probe the accessibility of pore-lining residues using Ag$^+$ ions, which have a similar radius to Na$^+$ and Ca$^{2+}$ ions (~1 Å) and should therefore permeate the TRPV1 pore. Unlike Na$^+$, however, Ag$^+$ ions form nearly irreversible complexes with cysteine residues and could obstruct the flow of current if the complex forms in the pore lumen (*del Camino and Yellen, 2001*). To accurately probe Ag$^+$-complex formation in the lumen of the pore, it is necessary to substitute a pore-facing residue with a cysteine in a background channel that lacks additional sites that could coordinate Ag$^+$ ions. We therefore began by testing a TRPV1 construct without native cysteines (Cys-less TRPV1) (*Salazar et al., 2008*; *Salazar et al., 2009*) and found that a low concentration of free Ag$^+$ (50 nM) reversibly inhibited whole-cell currents activated by two different TRPV1 ligands: 2-aminoethyldiphenyl borinate (2-APB) and capsaicin (*Figure 2A*). This suggests that Ag$^+$ either binds to a site on the external face of the channel to allosterically inhibit activation or binds within the pore to directly block permeation of Na$^+$. To distinguish between these two possibilities we measured the effect of transmembrane voltage, which is predicted to more strongly influence pore-block than allosteric inhibition because the electric field drops sharply across the pore (*Jiang et al., 2002*; *Contreras et al., 2010*). We measured the concentration-dependence of Ag$^+$ inhibition of Cys-less TRPV1 channels in the presence of a saturating concentration of capsaicin (10 μM) at several voltages (*Figure 2B–C and E*). The apparent affinity for Ag$^+$ decreased steeply with depolarization to positive membrane potentials (*Figure 2F*). The slope of this relationship (Zδ=−0.89) is equivalent to the transfer of a monovalent cation across 89% of the transmembrane electric field (*Woodhull, 1973*), strongly suggesting that Ag$^+$ ions block TRPV1 channels by binding within the pore, instead of allosterically inhibiting channel activation from outside the cation conduction pathway.

The high affinity of the blocking site for Ag$^+$ suggests it is located at the selectivity filter, where cations interact more intimately with the pore. If the filter is an activation gate, it would necessarily change conformation between open and closed states, possibly disrupting the blocking site for Ag$^+$ as channels close. This would result in a decrease in the apparent affinity for Ag$^+$ at lower agonist concentrations where channels are less active. In contrast, we observed a > 10 fold *increase* in the apparent affinity for Ag$^+$ (*Figure 2D,F and G*) when we measured the concentration-response relation for Ag$^+$ using a lower capsaicin concentration (50 nM) where the open probability of the channels was ~10 fold lower, as measured from conductance-voltage relations at the two capsaicin concentrations (*Figure 2H*). The prominent state-dependence of the apparent affinity of TRPV1 for Ag$^+$ ions, which are likely to interact with the filter based on the high affinity and voltage-dependence associated with block, strongly suggest that the filter undergoes a conformational change when channels activate. Although our findings indicate that Ag$^+$ ions bind more tightly to closed than to open channels at equilibrium, they provide no information about how gating influences Ag$^+$ accessibility. Interactions between Ag$^+$, Na$^+$, voltage, and distinct conformational states of the pore favored by capsaicin or voltage likely influence blocker affinity in complex ways. This might explain why we could not find a simple model for block that could explain the influence of voltage on

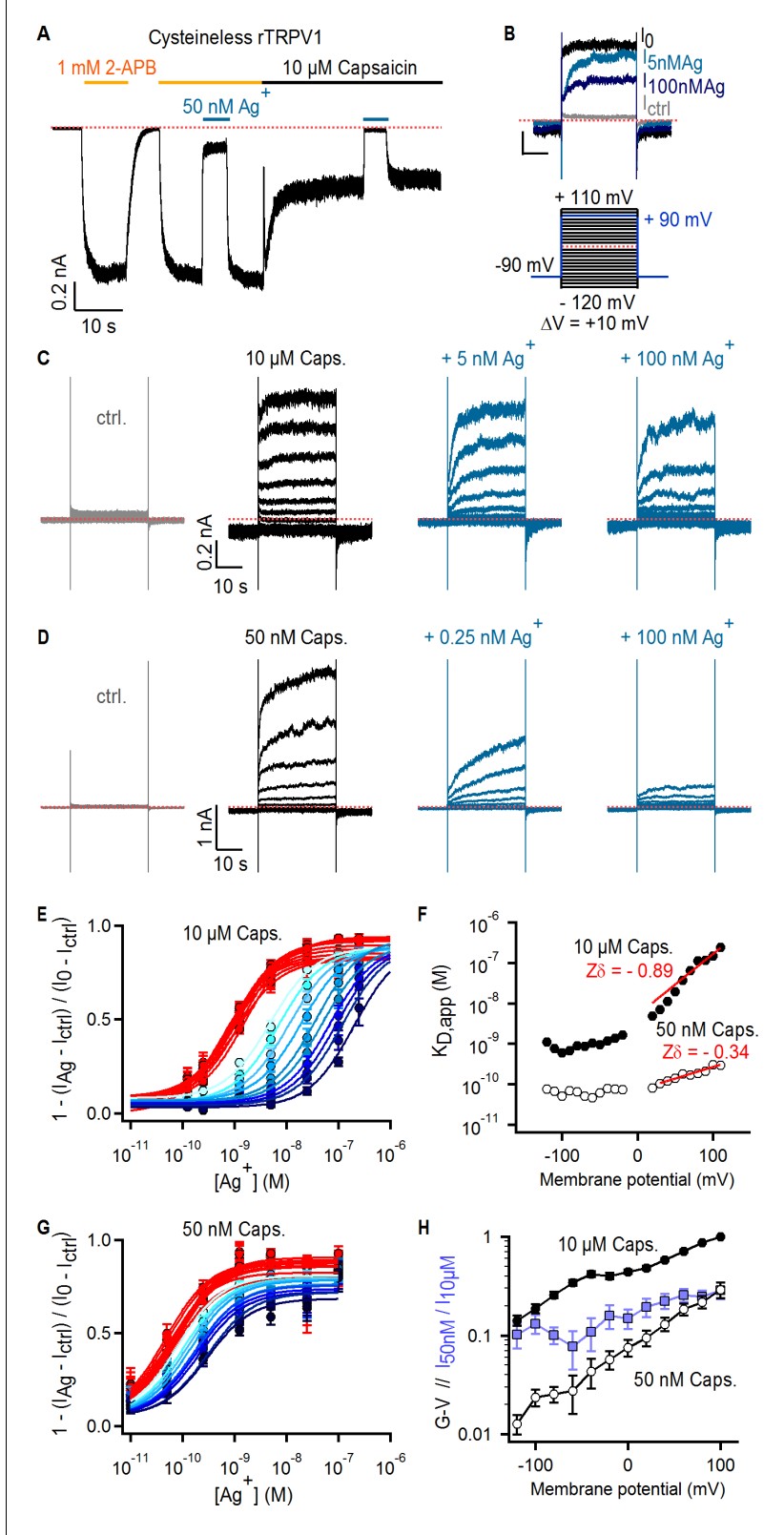

**Figure 2.** Ag[+] blocks TRPV1 channels in a state-dependent manner. (**A**) Representative whole-cell current-trace at −60 mV exposed to 2-APB (yellow line), capsaicin (black line) and external Ag[+]-ions (blue line). The red dotted line denotes the zero-current level. (**B**) Representative Cys-less rTRPV1 currents elicited by voltage steps from − 90 to + 90 mV (blue lines in the voltage protocol below) selected from the current families in (**C**), and obtained in the

*Figure 2 continued on next page*

*Figure 2 continued*

absence of agonist (gray trace), or with 10 µM capsaicin in the absence (black trace, $I_0$) or presence of external $Ag^+$ (blue traces). (**C, D**) Representative Cys-less rTRPV1 current traces elicited by voltage-steps as indicated in the protocol in (**B**, lower panel), activated with 10 µM (**C**) or 50 nM (**D**) capsaicin. Traces elicited at odd-voltage values are not depicted. (**E**) Dose-response relations for block by external $Ag^+$ at negative (red) or positive (blue, color intensity increases with depolarization) voltages, obtained from Cys-less rTRPV1 channels activated by 10 µM capsaicin from traces as in (**C**). Continuous curves are fits to the Hill equation. Data are shown as mean ± SEM (n = 5–10). (**F**) Apparent dissociation constants for $Ag^+$ as a function of voltage, obtained from fits of the Hill equation to each of the curves in (**E**) and (**G**). The red lines are fits of $K_{D,app}(V) = K_{D,app}(0\ mV) \times exp(-Z\delta \times VF/RT)$, with a $K_{D,app}(0\ mV)$ of 5.2 nM (10 µM capsaicin) or 70 pM (50 nM capsaicin). (**G**) Dose-response curves for $Ag^+$-block of currents activated by 50 nM capsaicin, obtained from current families as in (**D**) (mean ± SEM, n = 3–13). (**H**) Conductance-voltage (G–V) relations (black symbols) obtained from data as in (**C**) and normalized to 10 µM capsaicin at +100 mV. The ratio at each voltage between control-subtracted currents activated by 50 nM and 10 µM capsaicin is shown in blue. Data is shown as mean ± SEM (n = 5).

blocker affinity under all our experimental conditions, particularly at negative membrane potentials where blocker affinity appears to no longer be sensitive to voltage (*Figure 2F*).

## The selectivity filter of TRPV1 does not gate access to $Ag^+$

Encouraged by our observations that $Ag^+$ binds to both open and closed states of the pore, we set out to directly probe the state-dependence of $Ag^+$ accessibility across the selectivity filter of TRPV1. We introduced cysteine residues below the filter and measured how channel activation changes their accessibility to external $Ag^+$. Unlike the fully reversible block of Cys-less rTRPV1 by $Ag^+$, channels with inserted cysteines facing the pore are expected to be irreversibly inhibited by $Ag^+$. The fraction of $Ag^+$-dependent irreversible current inhibition provides a readout of the accessibility of the cation to the precise site where the cysteine residues have been introduced, if no other modifiable cysteines are present. We therefore used as background a Cys-less construct with a turret deletion (Δturret; residues 604–626) that boosts channel expression and lowers open probability ($P_o$) in the absence of agonists (*Jara-Oseguera et al., 2016*; *Geron et al., 2018*; *Dosey et al., 2019*). As we observed previously with Cys-less TRPV1 channels, $Ag^+$ reversibly inhibited Cys-less Δturret channels (*Figure 3—figure supplement 1A*); repeated stimulation of Cys-less Δturret channels with 2-APB resulted in a slow, progressive decrease in current (i.e. rundown), but the rundown measured from six initial rounds of stimulation with 2-APB without $Ag^+$ appeared unaffected by subsequent exposures to $Ag^+$ (*Figure 3C*, gray symbols), indicating that $Ag^+$ has no irreversible effects on our Cys-less Δturret background construct. In these experiments we used 2-APB instead of capsaicin as an agonist because it dissociates much faster from the channel when it is removed from the solution, which facilitates performing experiments where $Ag^+$ is applied in the absence of activators to probe $Ag^+$ accessibility through the filter in the closed state of the channel. Importantly, we verified that 2-APB does not directly interact with $Ag^+$ ions by showing that the presence of 2-APB does not alter the rate of modification by $Ag^+$ of a pore-facing cysteine in a P2X2 receptor construct (*Figure 3—figure supplement 2B C*) that was characterized previously (*Li et al., 2008*).

We first tested a cysteine introduced at I679; a position on the S6 helix that lies below the filter and facing the pore lumen (*Figure 1*). Cys-less Δturret + I679C channels had altered rectification (*Figure 3—figure supplement 3A–D*), but otherwise exhibited no drastic alterations in their responses to capsaicin or 2-APB (*Figure 3—figure supplement 2D–J*). In each accessibility experiment we repeatedly activated channels with a concentration of 2-APB that elicits near maximal $P_o$ (*Figure 3—figure supplement 2D F*). After six initial stimulations in the absence of $Ag^+$ to assess rundown (*Figure 3A*, left), cells were briefly exposed to extracellular $Ag^+$ during 2-APB application to assess accessibility of $Ag^+$ in the open state (*Figure 3A*, right). Exposure to $Ag^+$ rapidly inhibited all TRPV1-mediated current, as we observed previously in the background construct lacking pore-facing cysteines (Cys-less Δturret, *Figure 3—figure supplement 1A*), but in this case only a small fraction recovered upon removal of $Ag^+$ (*Figure 3A*, right). The irreversible inhibition by $Ag^+$ indicates that, when channels are open, external $Ag^+$ can traverse the filter and bind tightly to I679C, permanently obstructing $Na^+$ currents. We also found that the extent of rundown measured from the initial stimulations with 2-APB in the absence of $Ag^+$ was negligible compared with the rapid,

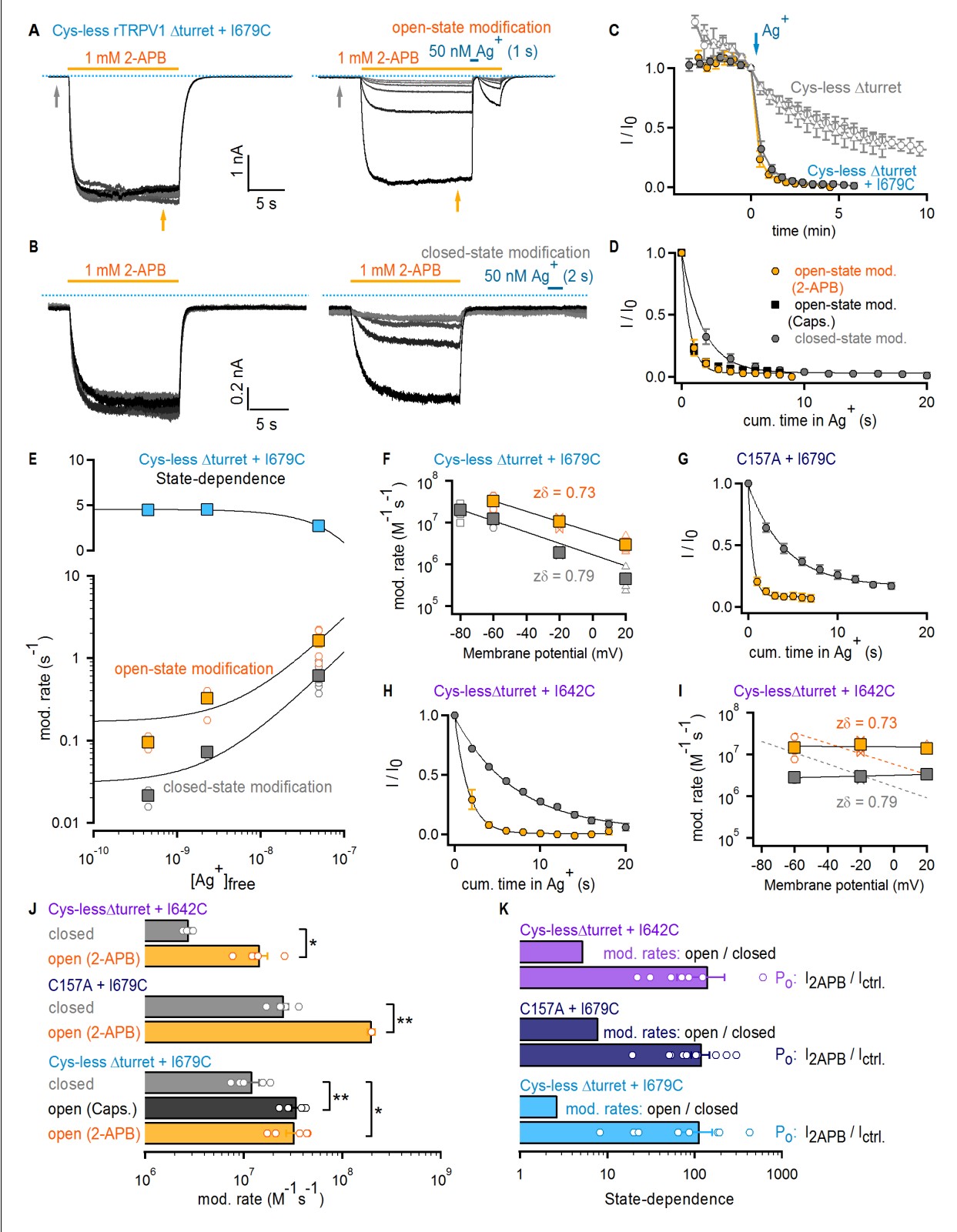

**Figure 3.** The selectivity filter of TRPV1 does not gate access to Ag+. (**A**) Current traces of Cys- less Δturret rTRPV1 + I679C channels from a representative Ag+-modification experiment in the open state at – 60 mV. The first six traces on the left panel were obtained without exposure to Ag+, followed by an exposure of 1 s duration per trace shown on the right panel. Only the first six traces with Ag+-exposure are shown. The grayscale-intensity of each trace decreases with repetition number, with the first trace in black. The dotted blue lines denote the zero-current level. (**B**)

*Figure 3 continued on next page*

*Figure 3 continued*

Representative current traces from a Ag$^+$-modification experiment in the closed state at – 60 mV, including the first six traces without exposure to Ag$^+$ on the left panel, and the first six traces with an exposure to Ag$^+$ of 2 s duration per trace on the right panel. (C) Mean Cys-less Δturret rTRPV1 + I679C time-courses for Ag$^+$-modification at −60 mV in the open (yellow) or closed (gray) states from experiments as in (A) and (B), respectively, and plotted as a function of total experiment time. To only quantify the irreversible component of Ag$^+$-inhibition, we plotted the steady-state currents in 2-APB before Ag$^+$-application for each trace (yellow arrow in panel (A), right). t = 0 was set at the first sweep with Ag$^+$-exposure (blue arrow) (mean ± SEM, n = 5), and was used to normalize each of the curves in the figure. The six data points before t = 0 correspond to the current values from the first recorded traces without exposure to Ag$^+$ (see left panels in [A] and [B] and *Figure 3—figure supplement 1A*), and reflect rundown caused by the repeated activation and de-activation of channels. The open gray symbols are data from experiments as in (A) for Cys-less Δturret rTRPV1 exposed to Ag$^+$ after t = 0 for 2 (triangles, n = 3) or 4 s (circles, n = 4) in the open state (see *Figure 3—figure supplement 1A*). (D) Mean time-courses of Ag$^+$-modification in the open (yellow, 2-APB; black, capsaicin – see *Figure 3—figure supplement 1B*) and closed (gray) states at −60 mV obtained by plotting the data in (C) after t = 0 as a function of cumulative time in Ag$^+$. Solid lines are fits to mono-exponential functions of time with parameters in (J). (E) Rates of modification in the open (yellow) and closed (gray) states obtained as in (A) and (B) at −60 mV and at different concentrations of free Ag$^+$ (0.45 nM, 2.3 nM and 50 nM). The black curves are fits to a line with a slope of 2.94 × 10$^7$ M$^{-1}$ s$^{-1}$(open state) or 1.16 × 10$^7$ M$^{-1}$ s$^{-1}$ (closed state). The colored squares are the mean ± SEM (n = 5), and the open circles are data from individual experiments. Representative traces for modification experiments at the two lower Ag$^+$ concentrations, as well as their corresponding time-courses of modification, are shown in *Figure 3—figure supplement 4*. The blue symbols in the upper panel are the ratios between the rates of modification in the open and closed states for each Ag$^+$-concentration, with a fit to a line shown in black. (F) Rates of Cys-less Δturret rTRPV1 + I679C modification by Ag$^+$ in the open (yellow) or closed (gray) states obtained from fits to data as in (D) at different voltages (see *Figure 3—figure supplement 1C–F*). The mean ± SEM are shown as squares (n = 5), and the rates from individual cells as open symbols. The black lines are fits to k(V)=k(0 mV) x exp(-zδ x VF/RT), with k(0)$_{open}$ = 5.8 x 10$^6$ M$^{-1}$s$^{-1}$; k(0)$_{closed}$ = 1.7 x 10$^6$ M$^{-1}$s$^{-1}$. (G) Mean time-courses of rTRPV1 C157A + I679C Ag$^+$-modification in the open (yellow) or closed (gray) states at −60 mV (see *Figure 3—figure supplement 1G,H* for representative traces and the time-courses including the initial data points without exposure to Ag$^+$) (mean ± SEM, n = 5). (H) Mean time-courses of Cys-less Δturret rTRPV1 + I642C Ag$^+$-modification in the open (yellow) or closed (gray) states at −60 mV (see *Figure 3—figure supplement 1I,J* for representative traces and the time-courses including the initial data points without exposure to Ag$^+$) (mean ± SEM, n = 5). (I) Rates of Cys-less Δturret rTRPV1 + I642C modification by Ag$^+$ in the open (yellow) and closed (gray) states obtained from fits to data as in (H) at different voltages (see *Figure 3—figure supplement 1I–K*). The mean ± SEM are shown as squares (n = 5), and the rates from individual cells as open symbols. Fits as in (F) are shown for data from I679C (dotted curves) or I642C (black curves): k(0)$_{open}$ = 1.5 x 10$^7$ M$^{-1}$s$^{-1}$; k(0)$_{closed}$ = 3.2 x 10$^6$ M$^{-1}$s$^{-1}$. (J) Mean modification rates at −60 mV (bars, mean ± SEM, n = 5). Rates from individual experiments are shown as open circles. The asterisks denote statistical significance as estimated from a heteroscedastic two-tailed t-test, with * for p<0.05 and ** for p<0.01. (K) Ratios between the current magnitude in the presence and absence of 2-APB (I$_{2APB}$/I$_{ctrl}$, see yellow and gray arrows in (A); data pooled from all modification experiments in the open and closed states at −60 mV), or between the rate of modification in the open and closed states at −60 mV.

The online version of this article includes the following figure supplement(s) for figure 3:

**Figure supplement 1.** State-dependence of Ag$^+$-accessibility in rTRPV1 channel mutants.

**Figure supplement 2.** Characterizing experimental conditions for Ag$^+$-modification experiments in cysteine-insertion constructs of TRPV1.

**Figure supplement 3.** Current-voltage relations for TRPV1 constructs used in the Ag$^+$-accessibility experiments.

**Figure supplement 4.** State-independent modification of Cys-less Δturret + I679C rTRPV1 channels at decreasing free Ag$^+$ concentrations.

**Figure supplement 5.** Lack of correlation between the rates of modification by Ag$^+$ in the closed state and the estimated open probability of cysteine-insertion constructs of TRPV1.

irreversible modification of I679C by Ag$^+$, as observed in time-courses obtained by plotting the steady-state current in 2-APB before (t < 0) and after (t ≥ 0) cells were exposed to Ag$^+$ during each stimulation with agonist (*Figure 3C*). If the selectivity filter functions as an activation gate, and assuming that the cumulative time-course of irreversible current inhibition (*Figure 3D*) reflects the rate with which external Ag$^+$ accesses I679C, then the rates of irreversible current inhibition by Ag$^+$ should depend on the open probability of the channels. By measuring the ratio of the current magnitude with and without agonist we estimated that the fold-change in P$_o$ resulting from channel activation by 2-APB is well above 100-fold (*Figure 3K*, light blue). If the selectivity filter is an activation gate, extracellular Ag$^+$ would not reach I679C in a closed channel. Therefore, if Ag$^+$ is applied extracellularly in the absence of 2-APB, the rate of cysteine modification should be >100 fold slower than in the presence of 2-APB. We performed these experiments (*Figure 3B*) and found that the time courses (*Figure 3C–D*) and corresponding rates of cysteine modification (*Figure 3J*) were very similar when Ag$^+$ was applied with or without 2-APB or capsaicin (<3 fold difference, *Figure 3K*, light blue). These results reveal that external Ag$^+$ has similar accessibility to I679C in both the open and closed states, strongly suggesting that the filter is not an activation gate.

As mentioned above, our interpretation of the Ag$^+$-modification experiments depends on the assumption that the rate of irreversible Ag$^+$-coordination is fast and therefore rate-limited by the accessibility of Ag$^+$ through the filter rather than by the modification reaction itself. In contrast, if

accessibility through the filter were several orders of magnitude faster than the modification reaction, it would be possible to obtain state-independent cysteine-modification rates even if the filter functions as an activation gate. Arguing against this possibility, the rate of irreversible inhibition by $Ag^+$ in the open state that we measured ($3.27 \pm 0.57 \times 10^7$ $M^{-1}s^{-1}$) is near the limit of diffusion, and falls within the same range as open-state modification rates by $Ag^+$ measured in other cation channels (*del Camino and Yellen, 2001*; *Flynn and Zagotta, 2001*; *Contreras et al., 2008*; *Li et al., 2008*). To further investigate whether the time-courses of cysteine modification by $Ag^+$ can reliably report on the state-dependence of $Ag^+$-accessibility through the filter, we performed $Ag^+$-modification experiments in the closed and open states using Cys-less Δturret + I679C channels and two considerably lower $Ag^+$-concentrations to effectively reduce the rate at which $Ag^+$ ions access the filter (*Figure 3—figure supplement 4A–D*). The rates of modification in the open state became measurably slower at the lower $Ag^+$-concentrations (*Figure 3E*, bottom panel, yellow), indicating that accessibility does indeed limit the rates of modification that we measure in our experiments. More importantly, the rates of modification in the closed state showed the same trend (*Figure 3E*, bottom panel, gray), resulting in a state-dependent ratio of open to closed-state modification that remained virtually constant over the entire range of $Ag^+$-concentrations (*Figure 3E*, top panel). These results demonstrate that our rates of modification closely report on $Ag^+$-accessibility, and that the lack of state-dependence that we observe demonstrates that the filter of TRPV1 does not function as an activation gate.

The rate of access of extracellular $Ag^+$ through the filter should also depend strongly on voltage. In order to determine whether membrane potential influences cysteine modification by $Ag^+$ ions, we carried out a series of experiments in which $Ag^+$ was applied with or without 2-APB at different membrane potentials (*Figure 3—figure supplement 1C–F*). The data reveal that negative membrane potentials have a strong positive influence on the rate of I679C modification by $Ag^+$ for both open and closed states (*Figure 3F*). This provides further evidence that $Ag^+$ reaches I679C in both the open and closed states of the channel by permeating through a large fraction of the transmembrane electric field (*Sobolevsky et al., 2005*; *Contreras et al., 2010*). To verify that our data aren't specific to the background construct or the location of the inserted cysteine, we performed experiments using either a full-length construct with all but one native cysteine known to influence gating upon reaction with intracellular cysteine-modifying agents (C157A + I679C) (*Salazar et al., 2008*; *Salazar et al., 2009*) (*Figure 3G* and *Figure 3—figure supplement 1G–H*) or a Cys-less Δturret construct with a cysteine introduced immediately below the filter at I642 (*Figure 1*, *Figure 3H*, and *Figure 3—figure supplement 1I–K*). In both cases we found that the rate of modification by $Ag^+$ was <10 fold slower in the absence than in the presence of 2-APB, whereas $P_o$ changed by >100 fold upon stimulation with 2-APB (*Figure 3J and K*). Importantly, Cys-less Δturret channels with the I642C mutation had normal rectification (*Figure 3—figure supplement 3E–F*), unlike those with the I679C mutation. However, in contrast with our data using I679C, the rates of modification for I642C were not influenced by voltage (*Figure 3I*). This suggests that the rate-limiting step in the time-course of modification is not the voltage-dependent access through the filter. However, the rates of modification for this position are <3 fold slower than those for I679C, so we think it is unlikely that the difference between the rates of access and of cysteine-modification in this construct is large enough to obscure any state-dependent control of accessibility through the filter. Finally, we found no consistent correlation between the rates of modification and the estimated change in $P_o$ by 2-APB in any of our closed state $Ag^+$-accessibility experiments (*Figure 3—figure supplement 5*), lending further support to the conclusion that the selectivity filter of the TRPV1 channel is not an activation gate.

## TRPV2 and TRPV3 do not gate access via the filter

Multiple stimuli, including cations (*Tominaga et al., 1998*; *Cao et al., 2014*; *Jara-Oseguera et al., 2016*) and toxins from venomous animals (*Bohlen et al., 2010*; *Cao et al., 2013*; *Yang et al., 2015b*), influence gating of TRPV1 by targeting the external pore. In contrast, no such modulators are known for its two close homologues, TRPV2 and TRPV3 (*Smith et al., 2002*; *Ryu et al., 2007*; *Gao et al., 2016a*). As discussed above for TRPV1, the selectivity filter of TRPV2 adopts non-conducting conformations in some of the available structures (*Figure 1—figure supplement 1A*) (*Zubcevic et al., 2016*; *Zubcevic et al., 2018b*; *Dosey et al., 2019*; *Zubcevic et al., 2019b*) and the filter has been suggested to function as a gate. In contrast, the selectivity filter of TRPV3 is more

solvent accessible in the absence of activators compared to TRPV1 and TRPV2 and has not been observed to undergo substantial structural changes in the presence of activators (*Singh et al., 2018*; *Zubcevic et al., 2018a*; *Zubcevic et al., 2019a*). We therefore thought it would be interesting to additionally examine the accessibility of external $Ag^+$ across the selectivity filters of TRPV2 and TRPV3 channels.

For our experiments with the TRPV2 channel, we used a rat TRPV2 construct (TRPV2-QM) containing four mutations at the S1-S4 domain (F472S, L507M, L510T, Q530E) that render it sensitive to the TRPV1-specific agonist resiniferatoxin (RTx), without otherwise altering its biophysical properties (*Yang et al., 2016*; *Zhang et al., 2016*). Although RTx is not useful for gated-accessibility experiments because the toxin dissociates extremely slowly and would interfere with measurements of closed-state modification, several structures of this mutant have been determined (*Zubcevic et al., 2018b*; *Zubcevic et al., 2019b*) and it has the advantage of less pronounced rundown compared to WT TRPV2 (data not shown). Experiments using 4 mM 2-APB as an activator confirmed that TRPV2-QM was reversibly blocked by $Ag^+$ in a similar way to Cys-less TRPV1 (*Figure 5—figure supplement 1A*), indicating that it constitutes a good background for our experiments because $Ag^+$ has no irreversible effects on this construct even after repeated exposures to $Ag^+$ (*Figure 4B and E*, gray triangles). To probe the state-dependence of accessibility for $Ag^+$ through the filter of TRPV2-QM, we introduced a cysteine at position I642 (*Figure 1—figure supplement 1A*), which aligns with position 679 in TRPV1. To our surprise, we found that TRPV2-QM + I642C channels exhibited a pronounced rundown upon repeated stimulation with 2-APB in the absence of $Ag^+$ (*Figure 4A and B*, empty circles). Because rundown in this construct was prominent enough to potentially obscure the effects of irreversible cysteine modification by $Ag^+$, we used the mean current time-course reflecting the rate of rundown in the absence of $Ag^+$ to compensate further recordings where we exposed cells to $Ag^+$ (see Materials and methods). We proceeded to perform accessibility experiments for rTRPV2-QM + I642C channels in which we applied $Ag^+$ either during or after 2-APB application (*Figure 4C*). In both cases, we found that irreversible cysteine modification by $Ag^+$ was more rapid than rundown, indicating that cysteine residues at I642C efficiently coordinate $Ag^+$ (*Figure 4D*). Importantly, the rundown-corrected time courses for irreversible cysteine modification by $Ag^+$ in the open or closed states (*Figure 4E*), and their associated rates of cysteine modification (*Figure 4F*), were less than 3-fold different (although this difference was indeed statistically significant). On the other hand, 2-APB increased $P_o$ by a factor >100 in these channels (*Figure 4G*), indicating that the filter is not an activation gate in TRPV2 channels.

To investigate $Ag^+$ accessibility in the TRPV3 channel, we initially tested whether $Ag^+$ has any effect on WT mTRPV3 channels and surprisingly discovered that unlike TRPV1 and TRPV2, $Ag^+$ is not an effective pore-blocker (*Figure 5—figure supplement 1B*). We therefore proceeded to introduce a cysteine at an equivalent site to that of I679 in TRPV1 (I674C, *Figure 1—figure supplement 1B*) using the WT mTRPV3 channel as background. As with TRPV1 and TRPV2, the time-courses of modification when $Ag^+$ was applied in the presence and absence of 2-APB (*Figure 5A–B* and *Figure 5—figure supplement 1C*) and the associated rates (*Figure 5D*) were very similar. However, we found that TRPV3 + I674C-expressing cells had baseline currents in the absence of 2-APB that were sensitive to $Ag^+$ (*Figure 5A*). When we plotted the time courses of $Ag^+$-mediated decay in baseline as a function of exposure time to $Ag^+$ (*Figure 5B*, insert), the rates were identical to the decay measured from the steady-state currents in 2-APB. This indicates that mTRPV3 + I674C channels contribute substantially to the baseline currents. Increasing the time interval between 2-APB removal and $Ag^+$ application did not alter the rates of modification (*Figure 5D* and *Figure 5—figure supplement 1C*), indicating that the increased baseline does not arise from incomplete deactivation after 2-APB removal, but rather from a higher baseline $P_o$. Regardless, the state-dependence of the rates of modification was smaller than the estimated change in $P_o$ by 2-APB (*Figure 5E*, dark blue), indicating that the filter of mTRPV3 is not an activation gate.

We wondered whether $Na^+$-ions bound within the pore of closed channels could obstruct access to $Ag^+$ and thus contribute to the small state-dependence of the rates of modification by $Ag^+$, which is statistically significant despite it being weak (*Figure 5D*). It is not possible to manipulate $Na^+$ occupancy of the pore of TRPV1 by removing the ion from the external solution because this manipulation activates the channel (*Jara-Oseguera et al., 2016*), precluding experiments at low $P_o$ and no external $Na^+$. However, removing external $Na^+$ does not activate TRPV3, making this TRPV channel an excellent choice for examining the extent to which $Na^+$ occupancy of the pore influences the

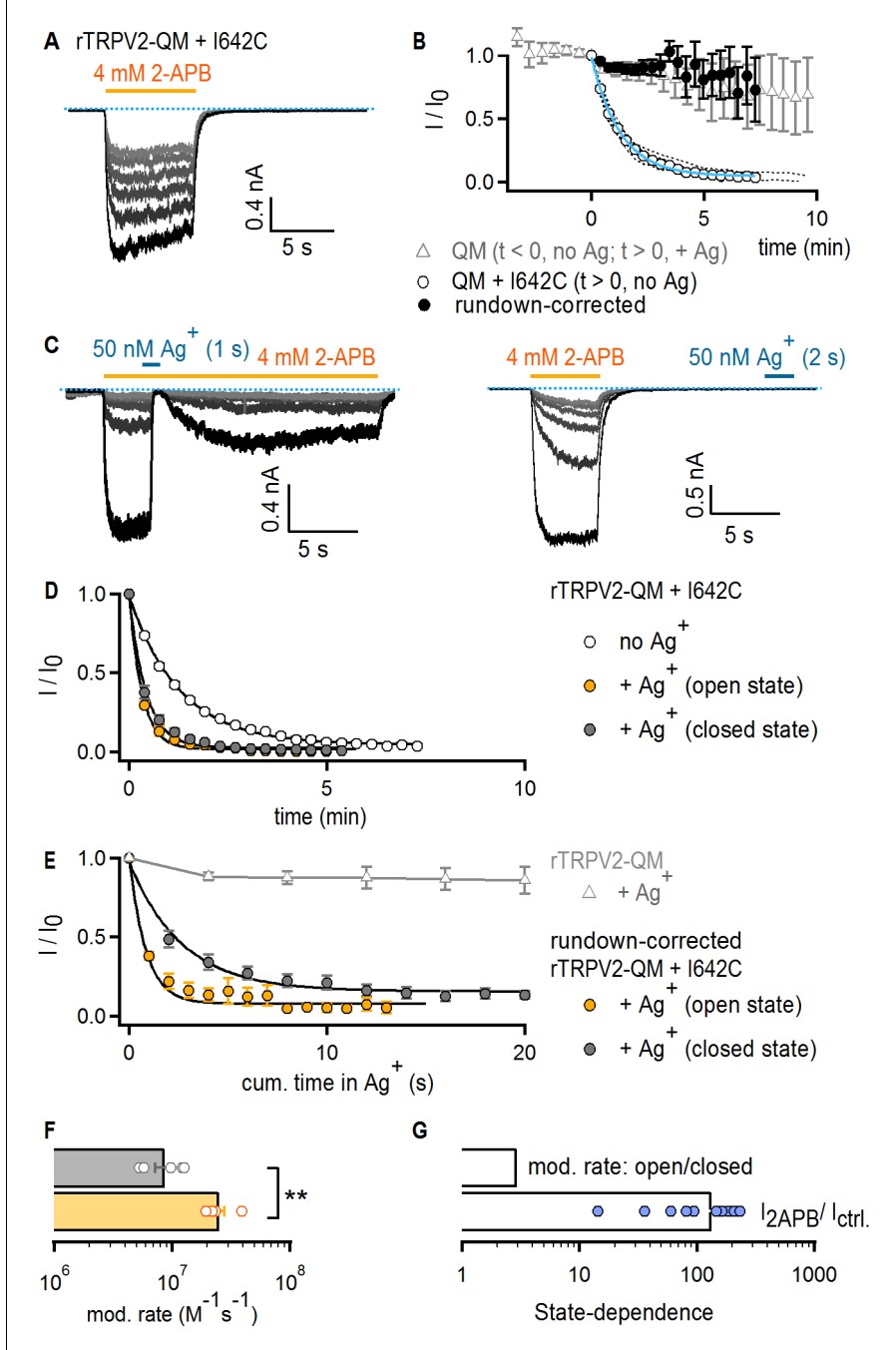

**Figure 4.** The TRPV2 channel does not gate access via the filter. (**A**) Representative current traces obtained by repeatedly activating channels with 2-APB (yellow horizontal line) at −60 mV without any exposure to Ag+ in a cell expressing rTRPV2-QM + I642C channels. The first recorded sweep is shown in black, with the subsequent five stimulations with agonist shown in decreasing grayscale-intensity. The blue dotted line denotes the zero-current level. (**B**) The empty circles are the time course of spontaneous rTRPV2-QM + I642C channel current decay (i.e. no Ag+-exposure) obtained from experiments as in (**A**) (mean ± SEM, n = 10; individual experiments are shown as dotted curves), and fit with: $I_{fit}(t) = (1-0.045) * exp(-t/84.57\ s) + 0.045$ (blue curve). The filled black symbols are the resulting mean time course for rTRPV2-QM + I642C (and no Ag+-exposure) after correcting each individual experiment for rundown using the exponential fit in blue (see Materials and methods). The time course for rTRPV2-QM (without any cysteine insertions) is shown in empty gray triangles (mean ± SEM, n = 9; see **Figure 5—figure supplement 1A** for representative current traces). For this construct, data at t < 0 correspond to the steady-state current values in 2-APB from the first six sweeps recorded without exposing cells to Ag+, whereas data at t > 0
*Figure 4 continued on next page*

*Figure 4 continued*

correspond to subsequently recorded sweeps in which cells were exposed to Ag$^+$ for 4 s per sweep in the presence of agonist. The data point at t = 0, right before the first exposure to Ag$^+$ was used for normalization. (C) Representative rTRPV2-QM + I642C current traces stimulated with 2-APB (yellow thick lines) and exposed to Ag$^+$ (blue thick lines) in the open (left panel) or closed (right panel) states at – 60 mV. The blue dotted line denotes the zero-current level. (D) Mean time-courses for Ag$^+$-modification in the open (yellow) or closed (gray) states (mean ± SEM, n = 5), together with the spontaneous current decrease measured from the same construct in experiments where cells are not exposed to Ag$^+$ (open circles, same data as in B). (E) Mean rundown-corrected time-courses for Ag$^+$-modification (see Materials and methods) in the open (yellow) and closed (gray) states as a function of cumulative time in Ag$^+$. The black curves are fits to mono-exponential functions, with parameters in (F). Empty gray triangles represent the data for TRPV2-QM + Ag$^+$ shown in (B) for t $\geq$ 0. (F) Mean rates of rTRPV2-QM + I642C modification in the open (yellow) and closed (gray) states obtained from mono-exponential fits to data as in (E). Bars are the mean ± SEM, with rates from individual experiments shown as open symbols (n = 5). The asterisks denote statistical significance as estimated from a heteroscedastic two-tailed t-test (p=0.001 < 0.01). (G) Ratios between the steady-state current magnitudes in the presence and absence of 2-APB (I$_{2APB}$ / I$_{ctrl}$; pooled from all modification experiments in the closed and open states), or between the rate of modification in the open and closed states as shown in (F).

rates of Ag$^+$ modification. We thus performed experiments with TRPV3 + I674C channels in which Ag$^+$ ions were applied in the absence of both 2-APB and external Na$^+$ (*Figure 5—figure supplement 1D*). We found that modification was detectably faster in the absence of external Na$^+$ ions, but not as fast as when 2-APB was also included (*Figure 5B and D*). This suggests that interactions between Ag$^+$ and Na$^+$ ions in the pore contribute to the observed state-dependence of Ag$^+$ accessibility.

Finally, we performed accessibility experiments with mTRPV3 channels with a cysteine introduced immediately below the filter at a site that aligns with position I642 in TRPV1 (I637C, *Figure 1—figure supplement 1B*). Unlike the I674C mutation, mTRPV3 + I637C channels did not exhibit baseline channel activity in the absence of 2-APB (*Figure 5—figure supplement 1E*). Nevertheless, the time courses of modification (*Figure 5C*) and the associated rates again showed marginal state-dependence (*Figure 5D and E*, purple). Together, our results indicate that the absence of a role for the selectivity filter in TRPV1 gating can be extended to TRPV2 and TRPV3 channels.

## Large organic cations permeate through the filter of TRPV1

The pore of TRPV1 channels has been suggested to 'dilate' upon prolonged activation, resulting in increased permeability to large organic cations (*Chung et al., 2008*; *Munns et al., 2015*). Structural changes in the selectivity filter are also thought to be necessary for permeation of large organic cations through TRPV2 channels (*Zubcevic et al., 2018b*; *Zubcevic et al., 2019b*). We therefore tested whether gating controls the access of extracellular organic cations to the pore of Cys-less rTRPV1 Δturret + I679C channels. We used two probes, MTSEA or MTSET, which modify cysteines with ethylamine or trimethyl-ethylamine adducts (*Figure 6—figure supplement 1C*), respectively. We first confirmed that our Cys-less Δturret background is not irreversibly affected by either reagent (*Figure 6—figure supplement 1A–C*). Furthermore, in these experiments we used capsaicin as an agonist because 2-APB appeared to react with the MTS reagents (data not shown). To estimate the accessibility of I679C to MTSEA in the activated state, we alternately stimulated channels with saturating concentrations of capsaicin in the absence (black traces) and presence (turquoise traces) of 2 mM MTSEA at a pH of 7.4 (*Figure 6A*, left). To test accessibility in the closed state, MTSEA was applied after removal of capsaicin (*Figure 6A*, right). Our protocol allowed us to independently assess rundown and current inhibition caused by cysteine-modification, in order to correct the time-courses for the effect of rundown (see Materials and methods). Prominent rundown-corrected MTSEA-dependent current inhibition was observed in both closed- (*Figure 6B*, gray symbols) and open-state experiments (*Figure 6B*, yellow symbols), indicating that MTSEA can gain access to I679C in both states of the channel. The time courses of modification by MTSEA (*Figure 6B*; gray and yellow) and the associated time constants (*Figure 6H*) were ~20 fold faster in the presence of capsaicin than in its absence. Yet, the change in P$_o$ caused by the agonist (>50 fold) was still larger than the decrease in the time constant of modification (τ) (*Figure 6I*, turquoise and white bars), suggesting that the filter does not fully gate access to MTSEA.

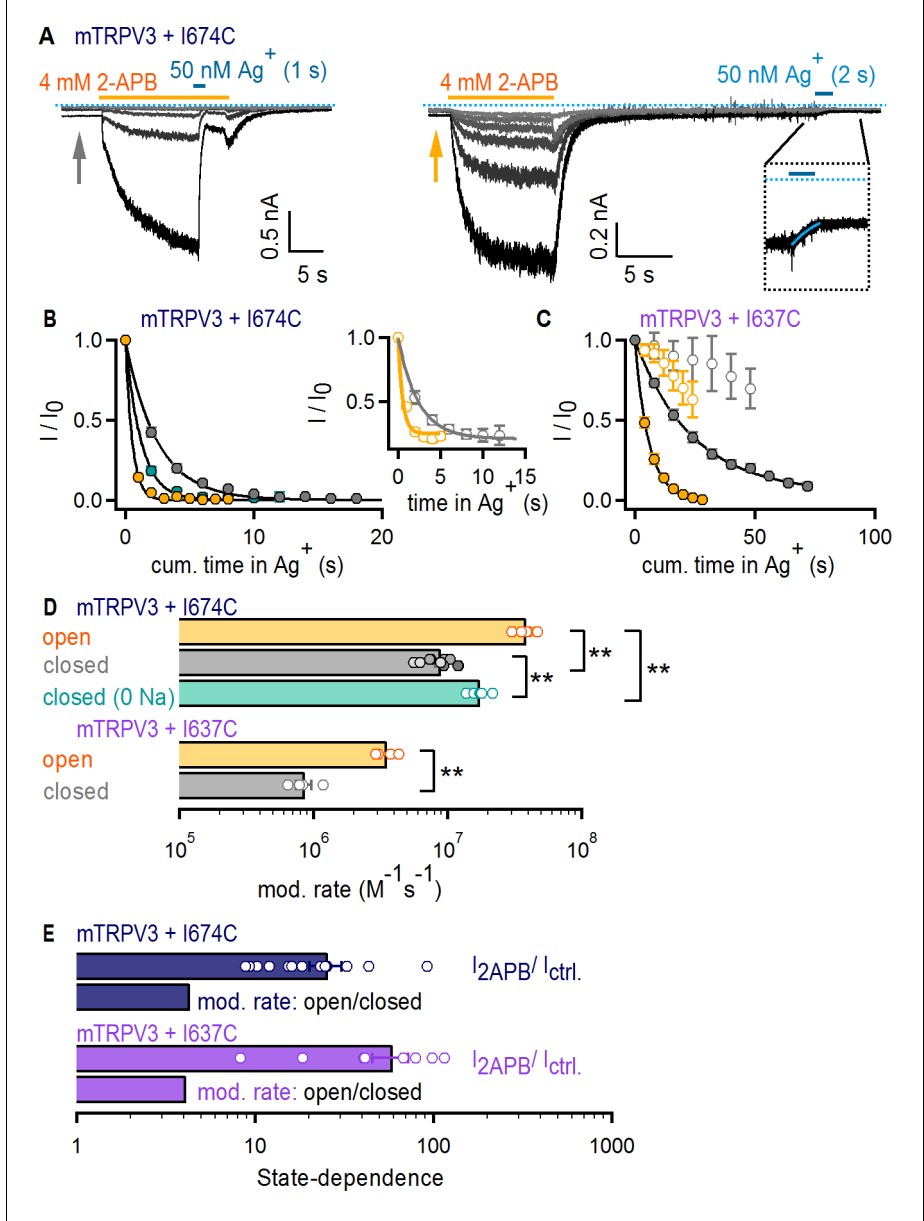

**Figure 5.** The filter does not function as an activation gate in the TRPV3 channel. (**A**) Current traces of mTRPV3 + I674C channels stimulated with 2-APB (yellow thick lines) from representative Ag$^+$-modification experiments in the open (left panel) or closed (right panel) states at – 60 mV. The zoomed view at the insert shows the Ag$^+$-dependent decrease in baseline current at the first sweep, fit with a mono-exponential function in blue. The blue dotted line denotes the zero-current level. (**B**) Mean time-courses for Ag$^+$-modification in the open (yellow) and closed (gray) states as a function of cumulative time in Ag$^+$ (mean ± SEM, n = 5–9; see **Figure 5—figure supplement 1C** for the time courses that include the first six initial recordings at t < 0 in each experiment, which were obtained without exposing cells to Ag$^+$). The turquoise symbols are data for Ag$^+$-modification in the closed state in the absence of external Na$^+$ (mean ± SEM, n = 6; see **Figure 5—figure supplement 1D** for representative current traces, and **Figure 5—figure supplement 1C** for the full time-course with data at t < 0). The black curves are fits to mono-exponential functions of time with associated rate-constants shown in (D, top panel). The insert shows the mean time courses for modification in the open (yellow) and closed (gray) states calculated from the baseline currents (gray and yellow arrows in A) (mean ± SEM, n = 7). (**C**) Mean time-courses for Ag$^+$-modification of mTRPV3 + I637C channels in the open (filled yellow symbols) and closed (gray filled symbols) states as a function of cumulative time in Ag$^+$ (mean ± SEM, n = 5). The black curves are fits to mono-exponential functions, with associated rate constants shown in (D) (see **Figure 5—figure supplement 1E** for representative current traces). The open symbols represent the initial six traces obtained without exposing cells to Ag$^+$ (**Figure 5—figure Figure 5 continued on next page**

*Figure 5 continued*
*supplement 1E*, left panel). (D) Modification rates for mTRPV3 channel mutants. The mean ± SEM are shown as bars (n = 5–9), with rates from individual cells shown as open circles. The asterisks denote statistical significance as estimated from a heteroscedastic two-tailed t-test, with ** for p<0.01. (E) Ratios between the steady-state current magnitude in the presence and absence of 2-APB ($I_{2APB}$ / $I_{ctrl.}$, pooled from all recordings in the closed and open states), or between the rates of modification in the open and closed states shown in (D).
The online version of this article includes the following figure supplement(s) for figure 5:

**Figure supplement 1.** State-dependence of $Ag^+$-accessibility in TRPV2 and TRPV3 channels with substituted cysteines.

Under our experimental conditions MTSEA exists mainly in a protonated, positively charged form. Our experiments therefore suggest that even large $MTSEA^+$ cations can traverse the filter when channels are closed. However, the non-protonated, neutral form of MTSEA ($MTSEA^0$), which co-exists at a ~ 1:10 ratio with the positively charged species at pH 7.4 (*Figure 6C*, gray circles) (*Holmgren et al., 1996*), could potentially access I679C through a non-aqueous pathway (e.g. through the membrane) without traversing the filter even though we included an excess of L-Cysteine (40 mM) in the pipette to act as MTSEA scavenger. If modification can be described as a simple bi-molecular reaction, its time constant should depend on the apparent pseudo first-order reaction rate constant ($k_{app}$) and the molar concentration of the reactive MTSEA species ($\tau$ = 1/($k_{app}$ x [MTSEA])). To determine which species reaches I679C in the closed state, we performed experiments in which MTSEA was applied at pH 10. At this pH, the concentration of $MTSEA^0$ increases by 10-fold and that of $MTSEA^+$ decreases by the same amount in comparison to pH 7.4 (*Figure 6C*, purple circles). We predicted that this would proportionally affect the time constant of modification and result in a ~ 10 fold *increase* if $MTSEA^+$ is the active species or a ~ 10 fold *decrease* if the active species is $MTSEA^0$. We observed an *increase* in $\tau$, but only by ~2 fold (*Figure 6B*, purple symbols, and H, purple), indicating a more complex reaction mechanism. We considered a model in which the pore is in equilibrium between two states that can bind MTSEA, but the cysteine is only accessible for modification in one state (*Figure 6D*). In simulated time courses of cysteine modification using this model, a decrease in the concentration of 10 mM MTSEA by 5-, 100-, 1 000- or 10 000-fold yielded increases in $\tau$ of only ~1, 3, 20- and 200-fold (*Figure 6E*, black curves). In contrast, a 10-fold shift in the conformational equilibrium towards the state where the cysteine is accessible decreased $\tau$ by exactly 10-fold (*Figure 6E*, yellow curve). To determine which MTSEA species is responsible for modification of I679C in the closed state, we collected data at pH 7.4 and ~10 using a lower concentration of MTSEA (*Figure 6B*, light blue and pink symbols). We then determined whether the calculated concentrations of either $MTSEA^+$ or $MTSEA^0$ in each of our four experimental conditions could account for the experimentally measured time constants of modification in the closed state. The trend in the experiments (*Figure 6B*; $\tau_{pink}$ > $\tau_{blue}$ > $\tau_{purple}$ > $\tau_{gray}$) correlates with the concentration of $MTSEA^+$ and not $MTSEA^0$ (*Figure 6F*, solid lines) and can also be well described by our model using concentrations for MTSEA that closely match those of $MTSEA^+$ in each of the experiments (*Figure 6F*, dotted curve). We thus conclude that $MTSEA^+$ can gain access through the filter when the channel is closed.

Finally, in accessibility experiments with the larger and positively charged MTSET (*Figure 6—figure supplement 1E*), we found slow accessibility in the open state and undetectable accessibility in the closed state that was indistinguishable from rundown (*Figure 6G* and *Figure 6—figure supplement 1F*). In addition, the ratio between the time constants of cysteine modification by MTSET in the closed vs the open state (*Figure 6H*) was for the first time noticeably larger than the fold-change in $P_o$ caused by capsaicin (*Figure 6I*, light blue and white bars). This indicates that the state dependence of accessibility across the selectivity filter to I679C increases with the size of permeant cations.

## Discussion

The biological functions of ion channels depend on their capacity to enable rapid conduction of ions across biological membranes in response to specific signals. Ion channels that share a transmembrane fold with TRP channels (*Yu et al., 2005*) appear to respond to their respective stimuli by

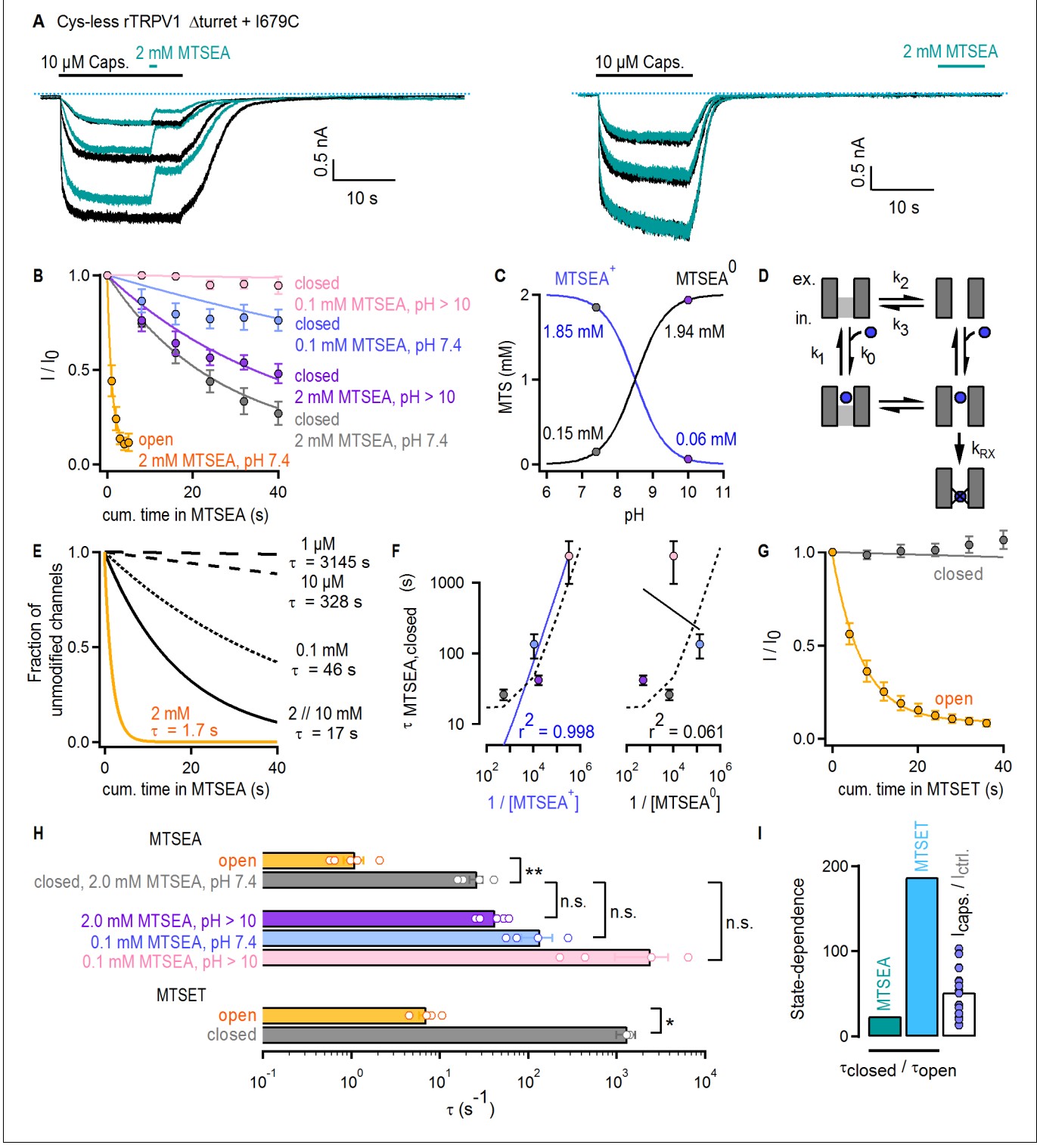

**Figure 6.** Large organic cations permeate through the filter of TRPV1. (**A**) Current traces of Cys-less Δturret rTRPV1 + I679C channels activated by capsaicin (thick black horizontal lines) from representative MTSEA-modification experiments in the open (left panel) or closed (right panel) states at – 60 mV. Recordings without (black traces) or with MTSEA exposure (turquoise traces) were acquired in an alternate fashion in the same experiment. The blue dotted lines denote the zero-current level. (**B**) Mean time-courses for MTSEA-modification as a function of cumulative time in MTSEA in the open (yellow) or closed states (mean ± SEM, n = 5 for 2 mM MTSEA; n = 4 for 0.1 mM MTSEA). All time-courses were corrected for rundown (see Materials and methods and *Figure 6—figure supplement 1D* for the time-courses depicting data before rundown correction and the time courses for rundown alone). The continuous curves are fits to mono-exponential functions of the form: $I_{norm}(t)=0.9 \times exp(-t/\tau) + 0.1$, with associated time-constants

*Figure 6 continued on next page*

*Figure 6 continued*

(τ) shown in (F) and (H). (C) Approximate concentrations of protonated (MTSEA⁺) and de-protonated (MTSEA⁰) MTSEA as a function of pH and a pK$_a$ = 8.5 (*Holmgren et al., 1996*) from 2 mM total MTSEA. The gray and purple circles mark the estimated concentration of each species at the pH in which experiments were done (either 7.4 or ~10). (D) Model for MTSEA-modification of TRPV1 channels. MTSEA (blue circle) binding and unbinding is given by rate constants k$_0$ x [MTSEA] and k$_1$, respectively. The equilibrium between two distinct channel states has rate constants k$_2$ and k$_3$. The rate of irreversible cysteine modification by MTSEA is given by k$_{RX}$. (E) Time-courses of modification by different MTSEA concentrations predicted by the model in (D), with parameters: k$_0$ = 0.5×10$^5$ M$^{-1}$s$^{-1}$ and k$_1$ = 50 s$^{-1}$; 'with agonist', k$_2$ = 0.6 s$^{-1}$ and k$_3$ = 10 s$^{-1}$ (yellow curve, [MTSEA]=2 mM); 'no agonist', k$_2$ = 0.06 s$^{-1}$ and k$_3$ = 10 s$^{-1}$ (black curves); k$_{RX}$ = 500 s$^{-1}$. MTSEA concentrations are shown to the right of the curves, together with the corresponding time-constant obtained from fits to a mono-exponential function of time as in (B). (F) Time-constants of modification in the closed state obtained from fits to data as in (B), plotted as a function of the inverse of the concentration of either of the two MTSEA species as calculated for each experimental condition (see Materials and methods) following the color-scheme in (B). The continuous curves are fits to a line, with the corresponding coefficient of determination (r$^2$) shown at the bottom of the graph. The dashed lines are the values from the model in (D and E). (G) Mean time-courses for MTSET-modification as a function of cumulative time in MTSET in the open (yellow) or closed (gray) states at −60 mV (mean ± SEM, n = 5; see *Figure 6—figure supplement 1E* for representative current traces). Data for modification in the closed state was corrected for rundown (see Materials and methods and *Figure 6—figure supplement 1F* for the time-courses depicting data before rundown correction and the time courses for rundown alone). The continuous curves are fits to mono-exponential functions of time, with associated time constants shown in (H). (H) Time-constants (τ) for MTSEA or MTSET modification obtained from fits to data as in (B) or (G). The bars are the mean ± SEM, and the open circles are data from individual experiments (n = 5 for 2 mM MTSEA and MTSET; n = 4 for 0.1 mM MTSEA). Statistical significance between different data-sets was assessed from heteroscedastic two-tailed t-tests, with n.s. (non-significant) for p>0.05, * for p<0.05 and ** for p<0.01. (I) Ratios between time-constants of MTSEA (2 mM, pH 7.4) or MTSET modification in the closed and open states, as shown in (H), or between the steady-state current magnitude in capsaicin vs control conditions (i.e. no agonist) (mean ± SEM, n = 16). Blue circles are data from individual cells.

The online version of this article includes the following figure supplement(s) for figure 6:

**Figure supplement 1.** MTSEA and MTSET modification of TRPV1 channels with substituted cysteines at position 679.

opening an S6 helix gate at the cytosolic side of the pore (*Liu et al., 1997*; *del Camino and Yellen, 2001*; *Shin et al., 2001*; *Rothberg et al., 2002*; *Xie et al., 2005*; *Oelstrom et al., 2014*; *Oelstrom and Chanda, 2016*) or a selectivity filter gate at the extracellular side of the pore (*Contreras and Holmgren, 2006*; *Wilkens and Aldrich, 2006*; *Contreras et al., 2008*; *Bagriantsev et al., 2011*; *Piechotta et al., 2011*; *Zhou et al., 2011*). TRP channels are less well characterized, but structural comparisons of different TRP channels in the presence and absence of activators have revealed constrictions in regions of the S6 helices and the selectivity filter that would prevent cation permeation (*Figure 1* and *Figure 1—figure supplements 1* and *2*). This has led to the hypothesis that TRP channels contain two activation gates – one at the filter and another at the cytosolic side – that must open in response to stimulation for ionic currents to flow. Here we establish that the selectivity filter of TRPV1, TRPV2 and TRPV3 channels permits access of Ag⁺ to sites deep within the pore in the absence of stimulation, indicating that it does not function as an activation gate. Moreover, because an organic cation as large as MTSEA⁺ can traverse the filter when TRPV1 channels are closed, the filter is unlikely to gate access to the physiological cations Na⁺ and Ca²⁺. Indeed, access of external Ag⁺ to cysteines on S6 likely involves displacement of Na⁺ ions within pore, which would require Na⁺ to exit the pore to the external side when the cytosolic gate is closed.

Our findings necessarily imply that TRPV1, TRPV2 and TRPV3 channels are gated by the S6 helices on the cytosolic side of the pore. This inference has strong support from cryo-electron microscopy and X-ray crystallography experiments with nanodiscs, amphipols and detergents (*Cao et al., 2013*; *Liao et al., 2013*; *Gao et al., 2016b*; *Huynh et al., 2016*; *Zubcevic et al., 2016*; *Singh et al., 2018*; *Zubcevic et al., 2018a*; *Zubcevic et al., 2018b*; *Dosey et al., 2019*; *Zubcevic et al., 2019a*; *Zubcevic et al., 2019b*). In addition, the majority of available TRP channel structures, which represent each of the TRP subfamilies, reveal constrictions at relatively conserved positions formed near the cytosolic ends of the S6 helices (*Figure 1* and *Figure 1—figure supplements 1* and *2*). This strongly suggests that the S6 helices function as the activation gate in all members of the TRP channel family. Consistently, the cytosolic constrictions formed by the S6 helices are wider in some structures obtained in the presence of activators. The presence of a cytosolic gate in the TRPV1 channel has additional experimental support from studies that explored the state-dependence of block by intracellular quaternary ammoniums (*Oseguera et al., 2007*; *Jara-Oseguera et al., 2008*), as well as the state-dependence of accessibility to the pore for intracellular cysteine-reactive probes (*Salazar et al., 2009*). This latter study scanned the S6 helix of TRPV1 for accessibility to cytosolic

$Ag^+$ and identified Y671 as the inner-most site in S6 that exhibits state-dependent modification (*Salazar et al., 2009*). Y671 is located above I679 – the site of the tightest constriction that was uncovered in the first closed state structures of TRPV1 (*Cao et al., 2013*; *Liao et al., 2013*; *Gao et al., 2016b*). Molecular dynamics simulations reveal the potential presence of an aqueous pathway between the S1-S4 and pore domain in TRPV1 through which $Ag^+$ ions could reach S6 cysteines located beyond the I679 gate (*Kasimova et al., 2018*), providing a possible explanation for this discrepancy. Additional experimental support exists for the presence of an activation gate formed by the S6 helices in other TRPV, TRPC, TRPM and TRPP channels (*Zheng et al., 2018a*; *Zheng et al., 2018b*). Interestingly, the sites of cytosolic constriction in most TRP channels are different to TRPV1, including those of TRPV2 (*Figure 1—figure supplement 1A*) (*Huynh et al., 2016*; *Zubcevic et al., 2016*; *Zubcevic et al., 2018b*; *Dosey et al., 2019*) and TRPV3 (*Figure 1—figure supplement 1B*) (*Singh et al., 2018*; *Zubcevic et al., 2018a*; *Zubcevic et al., 2019a*). Indeed, some TRP channels have up to two constrictions (*Figure 1—figure supplement 2*). The significance of these differences requires further investigation, but may originate from distinct helical breaks in S6 due to the disruption of backbone hydrogen bonds in so-called π-bulges (*Palovcak et al., 2015*; *Zubcevic et al., 2016*; *Kasimova et al., 2018*; *McGoldrick et al., 2018*).

At present it is unclear whether the selectivity filter could function as an activation gate in other TRP channels. So far, structures of TRP channels where the selectivity filter can be observed in a non-conducting conformation have only been obtained for TRPV1 (*Liao et al., 2013*; *Gao et al., 2016b*), TRPV2 (*Zubcevic et al., 2018b*; *Dosey et al., 2019*; *Zubcevic et al., 2019b*) and TRPP channels (*Shen et al., 2016*; *Grieben et al., 2017*; *Wilkes et al., 2017*; *Su et al., 2018*) (*Figure 1—figure supplement 2*). The selectivity filters of TRPML channels are narrow (*Figure 1—figure supplement 2*) but are thought to be more rigid than other members of the family owing to the extensive networks of interactions that hold them in place (*Hirschi et al., 2017*; *Schmiege et al., 2017*), suggesting little conformational flexibility and potentially precluding a role as activation gates. However, it is possible that subtle changes in conformation or dynamics could be sufficient to prevent cation permeation, as has been suggested for the inactivation gate in voltage-gated $K^+$-channels (*Pau et al., 2017*).

The agonist-induced structural differences in the filters of TRPV1 (*Cao et al., 2013*; *Gao et al., 2016b*) and TRPV2 (*Zubcevic et al., 2018b*; *Zubcevic et al., 2019b*) channel structures suggest that the external pore and the selectivity filter undergo activation-dependent conformational changes, in which 'wider' filter openings correspond to activated states. Moreover, mutations proposed to disrupt agonist-induced conformational changes in the filter of TRPV2 impair permeation of large organic cations without noticeably affecting $Na^+$ currents (*Zubcevic et al., 2018b*). The modulation of the single-channel current amplitude of the TRPV1 channel by extracellular protons (*Liu et al., 2009*; *Lee and Zheng, 2015*) and other cations (*Cao et al., 2014*), and the large state-dependence for block by $Ag^+$ that we detected in this study provide further support that the selectivity filter is a dynamic structure that can adopt multiple ligand-dependent conformations. Although the increased state-dependence in accessibility that we observe for MTSEA and MTSET relative to $Ag^+$ could originate from local state-dependent conformational changes in the S6 helices that alter reactivity of the introduced cysteines to these larger probes, they could also reflect activation-dependent conformational changes in the selectivity filter that result in increased accessibility of these large organic cations to sites deep within the pore. We therefore propose that the external pore and selectivity filter of TRPV1 are in a dynamic equilibrium between distinct conformations, and that channel activation favors those that enable more rapid permeation of large cations. Contrary to the 'pore-dilation' hypothesis (*Chung et al., 2008*; *Munns et al., 2015*), however, our data don't support the idea that permeability to large cations continues to increase once activation reaches equilibrium (within a few seconds). We observed no consistent evidence for slow increases in capsaicin- or 2-APB-elicited currents after channels had reached equilibrium. Moreover, even though different agonists result in distinct single-channel current amplitudes for TRPV1, the values remain stable during long recording periods (*Hui et al., 2003*; *Canul-Sánchez et al., 2018*; *Geron et al., 2018*). Also, the permeability of $Ca^{2+}$ relative to $Na^+$ in TRPV1 has been shown to remain stable during long agonist applications (*Samways et al., 2016*). P2X receptors are also permeable to organic cations, and observations originally interpreted as dilation of the pore were subsequently shown to result from accumulation or depletion of ions inside the cell (*Li et al., 2015*).

Our findings demonstrate that the stimulus-dependent conformational changes in the external pore and selectivity filter of TRPV1, if they occur under physiological conditions, do not function to limit the access of ions to the pore. To reconcile our findings with the structural and experimental observations of activation-dependent conformational changes in the selectivity filter and external pore region we propose that the selectivity filter functions as an actuator after adopting its active conformation to stabilize the cytosolic gate in an open conformation (*Figure 7*). This could constitute the mechanism by which the multiple stimuli that target the external pore of the TRPV1 channel promote opening of the cytosolic gate. These diverse stimuli include H⁺ (*Jordt et al., 2000*) and other cations (*Cao et al., 2014*; *Jara-Oseguera et al., 2016*), pain-producing toxins from poisonous animals (*Bohlen et al., 2010*; *Cao et al., 2013*; *Yang et al., 2015b*; *Bae et al., 2016*; *Gao et al., 2016b*), and possibly heat (*Grandl et al., 2010*; *Yang et al., 2015b*; *Jara-Oseguera et al., 2016*; *Zhang et al., 2018a*). Importantly, the mutual coupling between the cytosolic gate and the external pore and filter, and the principle of microscopic reversibility, would ensure that agonists like capsaicin and resiniferatoxin, which appear to induce opening of the cytosolic gate in TRPV1, TRPV2 and TRPV3 channels by acting on the S4-S5 linker helix (*Cao et al., 2013*; *Yang et al., 2015a*; *Gao et al., 2016b*; *Zubcevic et al., 2018b*; *Zhang et al., 2019*; *Zubcevic et al., 2019b*), also result in the stabilization of the selectivity filter region in its active conformation. The molecular mechanism of this coupling is not known, but in TRPV1 it has been suggested to involve interactions between T641 in the pore helix and Y671 in the S6 helix (*Steinberg et al., 2017*; *Kasimova et al., 2018*). A similar function has been suggested for the equivalent residues in TRPV2 (*Zubcevic et al., 2018b*; *Zubcevic et al., 2019b*). If the filter has a similar function in the TRPV2 or TRPV3 channels, what is its physiological significance? Unlike TRPV1, there are currently no known activators that act on the extracellular side of these channels, although they may yet be discovered. However, the external pore of TRPV1 has been implicated in temperature-sensing (*Grandl et al., 2008*; *Bae et al., 2016*; *Zhang et al., 2018a*), thus structural rearrangements in the selectivity filter region of TRPV2 and TRPV3 may allow this region to function as an actuator for heat activation.

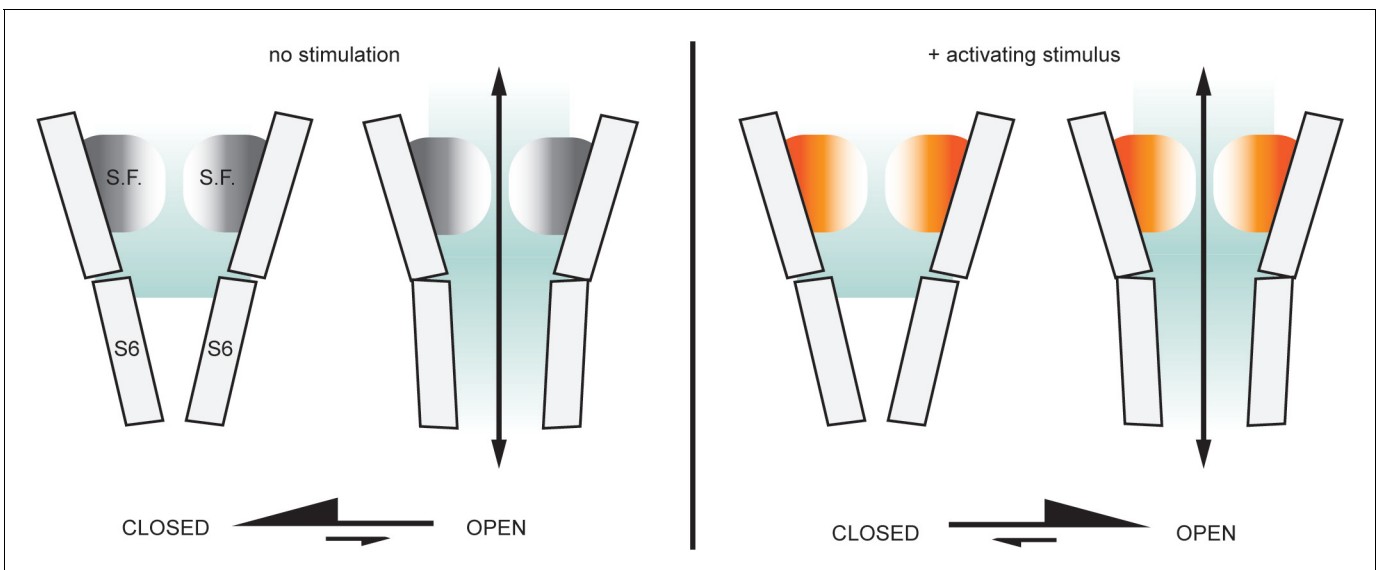

**Figure 7.** The external pore region could function as an actuator for TRP channel activation. In the absence of activating stimuli, the gating equilibrium of the TRPV1 channel is shifted towards the closed state; in most channels from a large ensemble, the S6 helix gate prevents permeation, whereas the selectivity filter (S.F.) is in a dynamic equilibrium between different conformations (gray/white shading) that still enable access of extracellular metal cations into the inner cavity. Upon stimulation, the conformational equilibrium of the outer pore shifts to favor a distinct set of conformations (orange/white shading) that may have a wider opening of the filter that facilitates access of large extracellular organic cations. We propose that this conformational shift within the selectivity filter and the external pore of the channel that occurs upon stimulation also contributes to stabilizing the open state of the S6 gate, effectively serving as an actuator for stimuli that interact with the extracellular surface of the channel.

# Materials and methods

## Cell culture

Human embryonic kidney cells (HEK293) were purchased from ATCC (CRL-1573), kept at 37°C in an atmosphere with 5% $CO_2$ and grown in Dulbecco's modified Eagle's medium (DMEM) supplemented with 10% fetal bovine serum (vol/vol) and 10 mg/mL gentamicin. For transfection, cells were detached with trypsin, re-suspended in DMEM and seeded onto glass coverslips in 3 mL dishes at 10–40% confluency. Transfections were performed on the same day using FuGENE6 Transfection Reagent (Roche Applied Science). TRP channel constructs were co-transfected with pGreen-Lantern (Invitrogen) at a ratio of 2:1 to visualize successfully transfected cells. Electrophysiological recordings were done 24–48 hr after transfection. Cells transfected with WT mTRPV3 or WT rTRPV1 were grown at 30°C to reduce expression levels.

## Molecular biology

The WT rat TRPV1 (*Caterina et al., 1997*), TRPV2 (*Caterina et al., 1999*) and P2X2 (*Brake et al., 1994*) channels were provided by Dr. David Julius (UCSF), and mouse TRPV3 (*Peier et al., 2002*) was provided by Dr. Feng Qin (SUNY Buffalo). The cysteine-less rat TRPV1 channel (*Salazar et al., 2008*) was provided by Dr. Tamara Rosenbaum (UNAM). This construct has all 18 native cysteine residues substituted by other amino acids: C21S, C31L, C63L, C73S, C126A, C157A, C257A, C362L, C386S, C390S, C442L, C578L, C616A, C621A, C634A, C715F, C741S and C766N. All constructs were cloned into modified pcDNA3.1(+) and pcDNA1 for high and low levels of expression, respectively. All mutations or deletions were introduced using the two-step PCR method, and verified by Sanger sequencing to check for PCR errors. We used a rat TRPV2 construct (TRPV2-QM) containing four mutations at the S1-S4 domain (F472S, L507M, L510T, Q530E) that render it sensitive to the TRPV1-specific agonist resiniferatoxin, without otherwise altering its biophysical properties (*Yang et al., 2016*; *Zhang et al., 2016*), and has been used for structural determination in several studies (*Zubcevic et al., 2018b*; *Zubcevic et al., 2019b*).

## Structural and sequence alignments

The TRP channel structures listed in the *Supplementary file 1* were obtained from the Protein Data Bank. The transmembrane (TM) domain of each structure was defined as the pre-S1, S1-S6 helices, and TRP box domains. The TM domains were aligned pairwise using Fr-TM-Align (*Pandit and Skolnick, 2008*), which aligns the structures and produces the corresponding sequence alignment based on structural similarity, rather than sequence identity. The pairwise alignments of each structure with the template channel (PDB ID: 6co7; nvTRPM2) (*Zhang et al., 2018c*) were merged to create the multiple sequence alignment, and template sequence gaps were removed. Pore geometry was determined using the MDAnalysis package implementation of the HOLE program (*Smart et al., 1993*; *Michaud-Agrawal et al., 2011*; *Gowers et al., 2016*), and the pore radius corresponding to each pore-facing residue was defined as the minimum pore radius constrained by any atom in the residue.

## Patch-clamp recordings

All experiments were done at room temperature (22°C) using the whole-cell configuration of the patch clamp, except for the noise-analysis which was done in inside-out patches. All electrophysiological data analysis was done using Igor Pro 6.34A (Wavemetrics). Data were acquired with an Axopatch 200B amplifier (Molecular Devices), filtered at 1 kHz for gap-free recordings (5 kHz for I-V relations and other protocols involving voltage-steps; 20 kHz for noise-analysis) and digitized at 4 kHz for gap-free recordings (20 kHz for protocols with voltage-steps; 100 kHz for noise analysis) with an Axon Digidata 1550A Low-Noise Data Acquisition System (Molecular Devices) and pClamp10 software (Molecular Devices). Pipettes were pulled from borosilicate glass (1.5 mm O.D. x 0.86 mm I.D. x 75 mm L; Harvard Apparatus) using a Sutter P-97 puller and heat-polished to final resistances between 0.5 and 5 MΩ using a MF-200 microforge (World Precision Instruments). 90% series resistance ($R_s$) compensation was used in all whole-cell recordings, and those cells with estimated voltage errors > 5 mV were discarded. An agar bridge (1M KCl; 4% wt/vol agar; teflon tubing) was used to connect the ground electrode chamber and the main recording chamber.

The intracellular recording solution consisted of (in mM): 130 NaCl, 10 HEPES, 10 EGTA, 10 MgCl$_2$, 60 sucrose, pH 7.4 (NaOH). MgCl$_2$ was included to both increase the quality of the seals and to block endogenous HEK293 channels. The intracellular solution for experiments with MTSEA was prepared daily and included 40 mM L-Cysteine as a scavenger (*Holmgren et al., 1996*) and 20 mM sucrose instead of 60 mM. The extracellular solution consisted of (in mM): 150 NaNO$_3$, 10 HEPES, 10 EDTA, 60 sucrose, pH 7.4 (NaOH). Although spontaneous disulfide formation between pore-facing cysteines can potentially confound the effects of Ag$^+$, we excluded reducing agents from all our experiments because we found that dithiothreitol (DTT) interfered with the kinetics of block by Ag$^+$ in Cys-less TRPV1 channels (*Figure 3—figure supplement 2A*). For most Ag$^+$-modification experiments we used 2-APB as agonist because it unbinds faster than capsaicin, and it is a common agonist of TRPV1, TRPV2 and TRPV3 channels (*Hu et al., 2004*). However, 2-APB is known to be fairly reactive (*Gao et al., 2016a*), so we verified that it did not interfere with a reported rate (*Li et al., 2008*) of Ag$^+$-accessibility to a pore-facing cysteine in P2X2 receptors (*Figure 3—figure supplement 2B C*). For experiments with MTSEA at pH ~10, NaOH was added directly to the MTSEA-containing recording solution and the pH was estimated using pH-indicator strips. For experiments with 10 mM MTSET, additional NaOH was also included to maintain pH at ~7.4 as estimated using pH-indicator strips. For both MTSEA and MTSET experiments, we used capsaicin as an agonist because 2-APB did apparently react with both cysteine-modifying molecules (data now shown). For experiments in the absence of external Na$^+$, identical solutions were used but Na$^+$ was entirely substituted with N-methyl-D glucamine (NMDG).

Capsaicin stock solutions (100 mM) were prepared in ethanol. Fresh 2-aminoethyldiphenyl borinate (2-APB) stock solutions were prepared daily at 1 M in DMSO, and vigorously shaken once added to recording solutions. The freshness of 2-APB seemed an important factor in our experiments because 'older' reagents seemed to result in more rundown, possibly because sub-maximal activation of TRPV1, caused by partial degradation of the agonist, seems to enhance rundown even in WT channels (personal observation, data not shown). We therefore used only 1 mg 2-APB packs and discarded them whenever prominent rundown was observed in the experiments. Fresh MTSEA and MTSET (Toronto Research Chemicals) stock solutions were prepared at 1 M in water and kept on ice and added to the recording solution right before each experiment. AgNO$_3$ stock solutions were prepared daily in water, first at 100 mM and then further diluted to 10 mM, for a final 1:1000 dilution for use in experiments at a total concentration of 10 µM in most experiments. Under our experimental conditions, a total of 10 µM AgNO$_3$ was calculated to yield 50 nM free Ag$^+$ using Max-Chelator Software. For experiments with 2.3 nM estimated free Ag$^+$, we started from a 100 mM stock solution, followed by a 1 mM stock and then a final total concentration in the recording solution of 0.4 µM. For experiments with 0.45 nM estimated free Ag$^+$, we prepared three stock solutions sequentially (100 mM, 1 mM, 0.2 mM) and used a final total concentration of AgNO$_3$ in the recording solution of 0.08 µM. All Ag$^+$-containing solutions were covered in aluminum foil to protect them from light, as well as the solution container in the perfusion system. All chemicals were from Sigma Aldrich unless stated otherwise.

A gravity-fed rapid solution exchange system (RSC-200, BioLogic) was used in all experiments. In each experiment, cells were lifted from the coverslip and placed in front of glass capillaries perfused with different solutions. For accessibility experiments in either the open or the closed states, the sequence of perfusion-line changes to and from Ag$^+$ was fixed to maintain identical conditions; for example, in experiments in the open state patches were first exposed to control solution (no agonist) in perfusion line 1, then to agonist in perfusion line 2, then to agonist + Ag$^+$ in perfusion line 3, followed by sequential return to lines two and then back to 1. For experiments in the closed state, control was at line 2, followed by agonist at line 1, then back to control at line 2, followed by Ag$^+$ (no agonist) at line 3, and then back to 2 and then 1. In all accessibility experiments, the timing for changes between perfusion lines was automated and controlled by the pClamp software and the motorized RSC-200 system.

The dose-response relations for Ag$^+$-block were obtained piece-wise due to the rectification of the channel and the voltage-dependence of block: cells with large-enough currents to adequately assess block at negative potentials had currents at positive potentials that were too big to clamp, and cells with currents that could be adequately clamped at positive potentials had currents at negative potentials that were too small and similar in magnitude as control currents in the absence of agonist. We therefore discarded data points from our current families where the estimated voltage

errors at positive potentials were >5 mV, or where the current amplitudes at negative potentials were too close to the control values in the absence of agonist. Data from several experiments were therefore pooled to construct the final dose-response curves. In addition, prolonged exposure to $Ag^+$ produced a progressive increase in leak currents, which further limited the accuracy of our recordings. We partially addressed this issue by repeatedly measuring I-V relations in capsaicin without $Ag^+$ throughout the experiment and normalized the currents in the presence of $Ag^+$ to the last-measured I-V curve in capsaicin alone. Unfortunately, due to the long times required to wash capsaicin off, we did not obtain leak currents except at the start of the experiment before the first application of capsaicin. In all I-V relations and protocols involving voltage-steps, these steps were applied at a frequency of 1 Hz.

For noise-analysis experiments, pipettes were covered in dental wax to reduce capacitive transients. In each experiment in the inside-out configuration, we obtained 50 traces of 400 ms duration at a constant voltage of −60 mV first in the absence of agonist and then in the presence of increasing concentrations of 2-APB or capsaicin. Patches were also held at −60 mV. Before acquiring the 50 test pulses, we made sure that the current had reached steady state at each agonist concentration.

## Dose-response relations for $Ag^+$-block

Dose-response curves were obtained by calculating the fraction of current blocked (F) for each $Ag^+$-concentration and membrane potential as: $F = 1- (I_{Ag} – I_{ctrl}) / (I_0 – I_{ctrl})$, where $I_{Ag}$ is the current magnitude at a given $Ag^+$ concentration and voltage, $I_{ctrl}$ is the initial current magnitude in the absence of agonist and $I_0$ is the current magnitude in the presence of agonist without $Ag^+$. Dose-response curves at every voltage were fit with the Hill equation with a fixed Hill coefficient (s) of 1.0: $F = base + \frac{\max - base}{1+\left(\frac{K_{D,app}}{[Ag]}\right)^s}$ . Unconstrained fits all yielded Hill coefficients close to 1 (not shown). We did not fix the max or base values in the fits, because both leak and maximal currents without $Ag^+$ were not perfectly constrained in our experiments.

## Conductance-voltage relations

To obtain the conductance-voltage (G-V) relations for TRPV1, we fit a second-order polynomial to data from I-V relations between −20 and +20 mV to calculate the reversal potential ($V_{rev}$), and then calculated the conductance from the I-V relations (excluding points from −20 to +20 mV) as: $G(V) = \frac{I_{steady-state}(V)}{V-V_{rev}}$.

## Noise analysis

Data for noise analysis was obtained in the inside-out configuration at several agonist concentrations in the same patch, either capsaicin or 2-APB, with 50 current traces of 400 ms duration per agonist concentration (2-APB: 0.05 mM, 0.1 mM, 0.25 mM, 0.5 mM, 1.0 mM, 4.0 mM; capsaicin: 20 nM, 50 nM, 200 nM, 1.0 μM, 10.0 μM, 20.0 μM; different sets of patches were used for capsaicin or 2-APB). The variance ($\sigma^2$) for each agonist concentration was calculated as: $\sigma^2(t) = \frac{1}{49}\sum_{j=1}^{50}\left(I_j(t) - \bar{I}(t)\right)^2$ (**Alvarez et al., 2002**; **Sigworth, 1980**), where $I_j(t)$ is the current for current trace number j at time t, and $\bar{I}(t)$ is the mean current for all 50 current traces at time t. We then calculated the steady-state $\sigma^2(t)$ and $\bar{I}(t)$ values as their mean at the end of the 400 ms trace, and obtained plots of $\sigma^2$ vs $\bar{I}$ for each individual patch and agonist concentration. The plots were fit to:

$$\sigma^2 = \bar{I} \times i - \frac{\bar{I}^2}{N}, \tag{1}$$

where i is the single-channel current amplitude at -60 mV, and N is the number of agonist-responsive TRPV1 channels in the patch. We calculated for each patch the open probability ($P_o$) for each agonist concentration as: $P_o = \frac{\bar{I}}{N\times i}$, and plotted the mean ± SEM for the $P_o$ and the normalized variance, from data pooled from all 5 patches per agonist. The magnitude of the single-channel current was slightly underestimated, as previously described for this analysis method (**Jara-Oseguera et al., 2016**). We used the $P_o$ values at each agonist concentration to also obtain agonist dose-response relations, which were fit with the Hill equation without any fitting constraints.

## Estimate for the change in $P_o$ in response to agonists

To estimate a lower limit for the change in $P_o$ in response to agonists, we simply took the ratio between the steady-state current magnitude in the presence of agonist and the basal currents in the absence of agonist, recorded before treatment with any cysteine modifier. This is expected to underestimate the true change in $P_o$, because leak currents constitute the major component for the current magnitude in the absence of agonist. In addition, the degree to which the change in $P_o$ caused by the agonists is underestimated by our method is very sensitive to the absolute magnitude of the current in the presence of agonist, because the level of leak was similar in most experiments, whereas current magnitudes in response to agonist could vary in over an order of magnitude between experiments. These issues with our estimate of $P_o$, however, do not diminish the strength our conclusions because for all experimental conditions and current magnitudes in the presence of agonist we found the change in $P_o$ estimated this way to always be larger than the change in the rates of modification in the open vs the closed state.

## Time-courses and rates of $Ag^+$ modification

For all representative current traces for cysteine-modification experiments, the zero-current level is indicated with a blue dotted line. For all $Ag^+$-modification experiments, we first obtained six current traces without any exposure to $Ag^+$, after which the metal was applied in either the open or the closed state, with a single application per trace. Some batches of cells exhibited more pronounced rundown, and we discarded all cells in which the 2-APB-activated current was reduced by more than 50% after the first six sweeps without $Ag^+$. Only the first six traces in which cells were exposed to $Ag^+$ are shown in the figures, with the first one at the front in black, and the gray-scale intensity decreasing with every repetition number. However, for most experiments we recorded a total of 12 traces with one exposure to $Ag^+$ in each. Time-courses for irreversible current inhibition caused by coordination of $Ag^+$ ions by introduced cysteines were obtained by measuring the mean steady-state current during stimulation with 2-APB and before exposure to $Ag^+$ in each sweep and plotted as a function of exposure time to $Ag^+$. In every experiment, we subtracted the control current in the absence of agonist measured from each current trace right before application of the agonist from the mean-steady state current in the presence of the agonist. This allowed us to correct for changes in baseline during the recording caused by an increase in the leak currents. We always used the mean steady-state current magnitude in the presence of agonist before the first application of $Ag^+$ in each experiment to normalize the entire time-course, which we also set as t = 0. Data at t < 0 correspond to the initial six traces recorded in each experiment before cells were exposed to $Ag^+$, and therefore rundown before t = 0 is observed as normalized current values > 1 in time courses where the entire experiment is depicted as a function of total experiment time. The rates of modification were calculated as $(1/\tau)/[Ag^+]_{free}$, where $\tau$ was obtained from the mono-exponential fits to the time-courses, and $[Ag^+]_{free}$ is the estimated concentration of free $Ag^+$. We did not apply any correction for rundown in most of our experiments because we considered its contribution to be negligible relative to the effects of $Ag^+$ under our experimental conditions. Importantly, we selected an overall $Ag^+$ concentration and exposure times to $Ag^+$ in each experiment that would result in much faster decrease in current due to modification by $Ag^+$ rather than rundown. We used the initial six traces without exposure to $Ag^+$ in every experiment to check that this was indeed the case. TRPV2-QM + I642C channels, however, exhibited more prominent rundown than other constructs and we therefore compensated for rundown in experiments with this construct. To do this we obtained a mean time course of current reduction upon repeated stimulations with 2-APB (and no exposure to $Ag^+$), fitted the data to a mono exponential function (*Figure 4B*, blue curve: $I_{fit}(t) = (1–0.045) * exp(-t/84.57 s) + 0.045$), and divided subsequent time courses starting at t = 0 by $I_{fit}(t)$ to compensate for rundown.

## Time-courses of modification by MTS reagents

The rates of irreversible covalent cysteine modification by the MTS reagents was in general slower than the irreversible effects of $Ag^+$, especially for MTSET and the lower concentration of MTSEA that we tested (0.1 mM). We therefore performed experiments in a way that would allow us to more accurately correct for the effects of rundown to obtain a better estimate of the actual rate of modification by the MTS reagents. To achieve this, we performed experiments in which cells were

alternatingly exposed to solutions that did or did not contain MTS reagents, while keeping everything else in the experiment identical, such as the duration of exposure to agonist or control solution without agonist. At the end of each experiment, which consistent of a total of 12 recorded traces, we obtained two separate time courses reflecting the decrease in current due to rundown, obtained from those traces in which the cell was not exposed to the MTS reagent, or the decrease caused by rundown in combination with the irreversible modification of cysteines by the MTS reagent, obtained from the traces in which the cell was exposed to the MTS reagent. To extract this information from the data, we first subtracted the control currents measured right before stimulation with the agonist from the steady-state currents in the presence of agonist (i.e. leak-subtraction), and from this we computed the fractional decrease in current $F$ between each pair of contiguous sweeps as: $F_j = (I_j - I_{j+1})/I_j$, where $I_j$ and $I_{j+1}$ are the mean leak-subtracted steady-state current magnitudes in capsaicin in two adjacent sweeps j and j+1. Because of the alternating protocol that we used, the calculated $F_j$ values alternatingly reflect the decrease due to rundown alone, or due to rundown together with the reaction with the MTS agent. Each data point in our rundown-corrected time courses was therefore obtained as: $I_{j+1} = I_j - I_j \times F_{2j+1} + I_j \times F_{2j}$ (see *Figure 6B and G*). The time-courses for rundown alone were obtained from: $I_{j+1} = I_j - I_j \times F_{2j}$ (see *Figure 6—figure supplement 1D and F*, empty symbols), and those including both rundown and MTS-dependent inhibition as: $I_{j+1} = I_j - I_j \times F_{2j+1}$ (see *Figure 6—figure supplement 1D and F*, filled symbols). We calculated [MTSEA$^+$] as a function of pH as: $[MTSEA]_{tot} \times [H^+] / (K_a + [H^+])$, and [MTSEA$^0$] as $[MTSEA]_{tot} - [MTSEA^+]$.

## Statistical analysis

All group data are shown as mean ± SEM. Some data were subject to a two-tailed heteroscedastic t-test as implement in Excel.

# Acknowledgements

We thank Shai Silberberg, Mufeng Li, Gilman Toombes, Antoniya Aleksandrova, Lucy R Forrest and members of the Swartz lab for helpful discussions. This work was supported by the Intramural Research Programs of the NINDS, NIH (to KJS) and by an NINDS Competitive Postdoctoral Fellowship and K99 Career Development Award (to AJO).

# Additional information

### Competing interests

Kenton J Swartz: Reviewing editor, *eLife*. The other authors declare that no competing interests exist.

### Funding

| Funder | Grant reference number | Author |
|---|---|---|
| National Institute of Neurological Disorders and Stroke | K99 Pathway to Independence Award | Andrés Jara-Oseguera |
| National Institute of Neurological Disorders and Stroke | Intramural Research Program NS002945 | Kenton J Swartz |
| National Institute of Neurological Disorders and Stroke | Competitive Postdoctoral Fellowship | Andrés Jara-Oseguera |

The funders had no role in study design, data collection and interpretation, or the decision to submit the work for publication.

### Author contributions

Andrés Jara-Oseguera, Conceptualization, Data curation, Software, Formal analysis, Supervision, Funding acquisition, Validation, Investigation, Visualization, Methodology, Writing—original draft, Project administration, Writing—review and editing; Katherine E Huffer, Software, Formal analysis,

Visualization, Methodology, Writing—review and editing; Kenton J Swartz, Resources, Supervision, Funding acquisition, Project administration, Writing—review and editing

#### Author ORCIDs
Andrés Jara-Oseguera (iD) https://orcid.org/0000-0001-5921-9320
Kenton J Swartz (iD) https://orcid.org/0000-0003-3419-0765

#### Decision letter and Author response
Decision letter https://doi.org/10.7554/eLife.51212.sa1
Author response https://doi.org/10.7554/eLife.51212.sa2

## Additional files

#### Supplementary files
• Supplementary file 1. TRP channel structures used in the structural alignment of the pore domain. All TRP channel structures included in the alignment are listed, indicating the PDB ID, the name of the channel, the DOI for the manuscripts where each structure is described, and the name of the species from which the proteins were cloned.

• Transparent reporting form

#### Data availability
All data generated or analysed during this study are shown in the manuscript and supporting files. The corresponding authors can be contacted if the raw data is required by anyone.

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
