## [Decision Letter]

**Acceptance summary:**

This study examines whether the selectivity filter acts as an activation gate in TRPV subfamily channels, which is an important open question in the entire TRP channel field. In several TRP channels, including TRPV1, structural data show clear opening, in response to agonists, of a gate near the cytosolic end of the transmembrane pore. However, the selectivity filter, located close to the extracellular end of the ion permeation pathway, also shows conformational changes upon agonist binding, and it is unclear whether under resting conditions it also restricts ion movement. Other 6-transmembrane tetrameric channels show variability in this respect; in some channels the selectivity filter serves as the gate, in others it does not. The authors test state-dependence of accessibility to extracellularly applied Ag^+^ ions of positions located intracellularly of the selectivity filter in TRPV1, TRPV2, and TRPV3 channels. They show that cysteines engineered into such positions are irreversibly modified by extracellularly applied Ag^+^, and find a minimal dependence of this modification rate on channel open probability. Even larger cationic reagents are seen to permeate through the selectivity filter of closed rTRPV1 channels. The conclusion is that the selectivity filter does not act as an activation gate in TRPV-type channels but is in a conducting conformation even when the channels are closed. On the other hand, the observed modest state dependence of modification rates implies some structural rearrangement in the filter coupled to gate opening. The work beautifully illustrates how rigorous functional studies can clarify molecular mechanisms in the post-structure era.

**Decision letter after peer review:**

Thank you for submitting your article "The selectivity filter functions as an actuator not a gate during TRPV1 channel gating" for consideration by *eLife*. Your article has been reviewed by three peer reviewers, including László Csanády as the Reviewing Editor and Reviewer #1, and the evaluation has been overseen by Richard Aldrich as the Senior Editor. The following individuals involved in review of your submission have agreed to reveal their identity: Seok-Yong Lee (Reviewer #2); Tibor Rohacs (Reviewer #3).

The reviewers have discussed the reviews with one another and the Reviewing Editor has drafted this decision to help you prepare a revised submission.

Summary:

This exhaustive study examines whether the selectivity filter acts as an activation gate in TRPV subfamily channels, which is an important open question in the TRP channel field. Structural data show clear opening of the cytoplasmic S6 helix gate in response to agonists in several TRP channels, including TRPV1. The selectivity filter however also shows conformational changes upon agonist binding, and it is not clear whether under resting conditions it restricts ion movement. Other six transmembrane tetrameric channels show variability in this respect; in some channels the selectivity filter serves as the gate, in others it does not. The authors test state-dependence of accessibility to extracellularly applied Ag^+^ ions of two different positions located intracellularly of the TRPV1 selectivity filter. They show that cysteines engineered into positions 679 or 642 of rTRPV1 are irreversibly modified by extracellularly applied Ag^+^, and find a minimal dependence of this modification rate on channel open probability. They then obtain similar results on rTRPV2 and mTRPV3 channels. Finally, they show that even larger cationic reagents such as MTSEA can permeate through the selectivity filter of closed rTRPV1 channels and modify the engineered cysteine at position 679. The conclusion is that the selectivity filter does not act as an activation gate in TRPV-type channels but is in a conducting conformation even when the channels are closed. The data are convincing and support the above conclusion. However, several concerns were raised by the reviewers regarding presentation and interpretation which will need to be addressed.

Essential revisions:

1) The authors compare the fold-change in P_o_ with the fold-change in modification rate by Ag^+^ upon agonist stimulation. They consistently observe state-dependence for the modification rate but smaller than the fold-change in P_o_ (2-10 fold vs. >100 fold), which forms the basis of the authors' conclusion that the selectivity filter does not act as a gate. However, this comparison assumes that the rate of covalent modification of the cysteine by Ag^+^ is fast, and therefore rate limited by Ag^+^ permeation through the selectivity filter. Thus, comparison of the change in modification rate with the fold-change in Po is not an ideal control. The target position 679 (in rTRPV1) is below the filter but above (or right at) the cytosolic gate, and the authors claim that it is accessible to extracellular Ag^+^ in both open and closed channels. A better control for this claim would be to engineer a cysteine into a target position which is below the cytosolic gate (e.g., G683C or N687C), and test its accessibility to external Ag^+^. Demonstration of pronounced state dependence of accessibility would provide a strong control for the experiments targeting position 679, and prove that extracellular Ag^+^ can access the engineered pore cysteines indeed only by entering through the pore itself. Furthermore, this would allow the authors to state not only that the filter is not a gate, but also that the bundle crossing is a gate.

2) The authors interpret the modest state-dependence of accessibility of position 679 to external Ag^+^ and MTSEA^+^, and its inaccessibility to MTSET^+^ in the closed state, to imply that the selectivity filter changes conformation upon activation (Discussion, third paragraph). Instead, accessibility might be restricted locally, in the immediate vicinity of position 679. Indeed, the choice of position 679 for showing external accessibility through the pore in the closed state is a bit unfortunate, as I679 forms the cytoplasmic gate, which itself obviously undergoes state-dependent conformational changes (it constricts in the closed state). The other tested position, 642, is so close to the filter that again alternative explanations may come into play. Demonstration of a modest state dependence of accessibility of cavity position 675 would be a more convincing argument for supporting a state-dependent conformational change of the filter. The authors should either add such experiments or tone down the conclusion about state-dependent conformational changes in the filter, leaving room for alternative explanations.

3) While the data show convincingly that the selectivity filter does not serve as a gate, we do not think the authors made the case that it serves as an actuator. Indeed it is not entirely clear what they mean by this term. There is little evidence to be found in the data to support the notion that the selectivity filter is able to regulate opening of the cytoplasmic gate as none of the agonists used in the study (2-APB, capsaicin) act on the external pore. Without experiments that directly probe stimuli that act on the external part of the pore (DkTx or H^+^) it is difficult to make conclusions about the ability of the selectivity filter to regulate the opening of the cytoplasmic gate. We recommend removing speculations on the "actuator" role of the filter from title and Abstract, and addressing this possibility more carefully in the Discussion.

4) Extrapolation of the authors' conclusions on TRPV1, TRPV2 and TRPV3 to other TRP channels seems somewhat problematic. First, TRP channels exhibit a range of ionic selectivities (Ca^2+^ selective TRPV5 and TRPV6, monovalent selective TRPM4 and TRPM5) and structures around the external pore (two pore helices in TRPA1, the tight structure of the selectivity filter in TRPML3, the wide-set external pore of TRPM8). Even within the TRPV1-4 subfamily, there are significant differences in the structure of the external pore. For example, TRPV3 does not possess a turret like TRPV2 and TRPV1 and its selectivity filter appears not to change during ligand gating (Singh et al., 2018; Zubcevic et al., 2018; Zubcevic et al., 2019). In light of such variability, extrapolation of the present results to other TRP channels without supporting experimental data is speculative.

5) The manuscript could benefit from more clarity in the writing and better explanations for the use of specific constructs and agonists. E.g. why do the authors use 2-APB in some experiments and capsaicin in others? Why use the full-length C157A construct rather than wild type TRPV1? Why is the TRPV2-QM construct (please define construct!) used when channels are activated only by 2-APB? Why wasn't RTx used in any of the experiments? As previous work from the authors has shown, it is a potent agonist for both TRPV1 and TRPV2, with P_o_ values approaching 1.

---

## [Author Response]

Essential revisions:1) The authors compare the fold-change in Po with the fold-change in modification rate by Ag+ upon agonist stimulation. They consistently observe state-dependence for the modification rate but smaller than the fold-change in Po (2-10 fold vs. >100 fold), which forms the basis of the authors' conclusion that the selectivity filter does not act as a gate. However, this comparison assumes that the rate of covalent modification of the cysteine by Ag+ is fast, and therefore rate limited by Ag+ permeation through the selectivity filter. Thus, comparison of the change in modification rate with the fold-change in Po is not an ideal control. The target position 679 (in rTRPV1) is below the filter but above (or right at) the cytosolic gate, and the authors claim that it is accessible to extracellular Ag+ in both open and closed channels. A better control for this claim would be to engineer a cysteine into a target position which is below the cytosolic gate (e.g., G683C or N687C), and test its accessibility to external Ag+. Demonstration of pronounced state dependence of accessibility would provide a strong control for the experiments targeting position 679, and prove that extracellular Ag+ can access the engineered pore cysteines indeed only by entering through the pore itself. Furthermore, this would allow the authors to state not only that the filter is not a gate, but also that the bundle crossing is a gate.

We agree with the reviewers that the central assumption that the rate of cysteine modification by Ag^+^ was not formally tested in any of our experiments. We attempted to perform modification experiments using a construct with a substituted cysteine below the I679 gate, but none of the constructs we generated (L681C, G683C and N687C) were irreversibly modified by Ag^+^ (although all of them exhibited Ag^+^-dependent block like that observed in the cysteine-less background construct). Instead, we have now included data showing that the ratio between the rates modification in the closed and open states is maintained over a wide range of Ag^+^-concentrations below the one we had tested in the original version of the manuscript (Figure 3E and Figure 3—figure supplement 4). Notably, the modification rates in the closed and open state become slower as the concentration of Ag^+^ is decreased, indicating that our rates reflect accessibility of the metal. We have also included a paragraph discussing this issue (subsection “The selectivity filter of TRPV1 does not gate access to Ag^+^”, third paragraph). As a result of the inclusion of the new data, we have moved the data showing a lack of correlation between the rates of modification in the closed state and our estimate of Po to the new Figure 3—figure supplement 5.

2) The authors interpret the modest state-dependence of accessibility of position 679 to external Ag^+^ and MTSEA^+^, and its inaccessibility to MTSET^+^ in the closed state, to imply that the selectivity filter changes conformation upon activation (Discussion, third paragraph). Instead, accessibility might be restricted locally, in the immediate vicinity of position 679. Indeed, the choice of position 679 for showing external accessibility through the pore in the closed state is a bit unfortunate, as I679 forms the cytoplasmic gate, which itself obviously undergoes state-dependent conformational changes (it constricts in the closed state). The other tested position, 642, is so close to the filter that again alternative explanations may come into play. Demonstration of a modest state dependence of accessibility of cavity position 675 would be a more convincing argument for supporting a state-dependent conformational change of the filter. The authors should either add such experiments or tone down the conclusion about state-dependent conformational changes in the filter, leaving room for alternative explanations.

The reviewers are right to point out that our data with MTSEA and MTSET can be explained by local state-dependent changes in the S6 rather than by conformational changes in the filter, which we did not stress clearly enough in the previous version of the manuscript. We have now made this clear in the Discussion section (Discussion, fourth paragraph). However, we believe that both the data from the structures, as well as several lines of evidence from functional experiments, strongly suggest that conformational changes in the filter do occur as channels activate, at least in the case of TRPV1 and TRPV2. We have therefore re-organized our discussion of our findings to reflect that our MTSEA and MTSET are consistent with the data from previous reports, although they still might have alternative explanations.

3) While the data show convincingly that the selectivity filter does not serve as a gate, we do not think the authors made the case that it serves as an actuator. Indeed it is not entirely clear what they mean by this term. There is little evidence to be found in the data to support the notion that the selectivity filter is able to regulate opening of the cytoplasmic gate as none of the agonists used in the study (2-APB, capsaicin) act on the external pore. Without experiments that directly probe stimuli that act on the external part of the pore (DkTx or H^+^) it is difficult to make conclusions about the ability of the selectivity filter to regulate the opening of the cytoplasmic gate. We recommend removing speculations on the "actuator" role of the filter from title and Abstract, and addressing this possibility more carefully in the Discussion.

We have now removed the speculations on the actuator mechanism from the title and presented them in the Abstract in a clearer fashion and as a proposal that does not derive directly from the findings in this manuscript. We agree that there is no evidence in our data in support of this mechanism. However, we think that the evidence from both the structural and functional data strongly suggests that the selectivity filter region does indeed undergo conformational changes that are associated with channel activation. We therefore think that it is important to provide a hypothesis to reconcile the presence of this conformational changes with our finding that the selectivity filter does not function as an activation gate. We have now more clearly described this hypothesis in the last paragraph of the Discussion.

4) Extrapolation of the authors' conclusions on TRPV1, TRPV2 and TRPV3 to other TRP channels seems somewhat problematic. First, TRP channels exhibit a range of ionic selectivities (Ca^2+^ selective TRPV5 and TRPV6, monovalent selective TRPM4 and TRPM5) and structures around the external pore (two pore helices in TRPA1, the tight structure of the selectivity filter in TRPML3, the wide-set external pore of TRPM8). Even within the TRPV1-4 subfamily, there are significant differences in the structure of the external pore. For example, TRPV3 does not possess a turret like TRPV2 and TRPV1 and its selectivity filter appears not to change during ligand gating (Singh et al., 2018; Zubcevic et al., 2018; Zubcevic et al., 2019). In light of such variability, extrapolation of the present results to other TRP channels without supporting experimental data is speculative.

We agree with the reviewers and have removed the statement claiming that our findings related to the role of the selectivity filter in gating in TRPV1, TRPV2 and TRPV3 could extend to other members of the TRP family, and substituted it with a new paragraph that simply states that for most other TRP channels, the filter has not been observed to adopt non-conducting conformations (Discussion, third paragraph). We still present the argument that the S6 helix is likely to form the activation gate for all members of the TRP family, as we think there is enough evidence from the structures and from some functional experiments to support this hypothesis (Discussion, second paragraph).

5) The manuscript could benefit from more clarity in the writing and better explanations for the use of specific constructs and agonists. E.g. why do the authors use 2-APB in some experiments and capsaicin in others? Why use the full-length C157A construct rather than wild type TRPV1? Why is the TRPV2-QM construct (please define construct!) used when channels are activated only by 2-APB? Why wasn't RTx used in any of the experiments? As previous work from the authors has shown, it is a potent agonist for both TRPV1 and TRPV2, with Po values approaching 1.

We thank the reviewers for pointing this out. We have now included additional information on the use of agonists and the selection of constructs in the Results section. We used 2-APB in most experiments because it maximally activates TRPV1, TRPV2 and TRPV3 channels with a dissociation constant that is much faster than capsaicin or RTx, which was important for the experiments in the closed state. 2-APB appeared to react with the MTS reagents, and therefore we used capsaicin as an agonist in these experiments. We selected TRPV2-QM for our experiments because this construct exhibits less rundown compared to the WT channel and has been used in many of the structural studies. We do not use RTx because it has exceedingly high affinity (and accumulates in membranes) and does not appear to dissociate from the channels once they are activated, precluding closed-state accessibility experiments. Finally, we used the C157A mutant as C157 has been shown to react with intracellular cysteine-modifying reagents. Importantly, the C157A mutation behaves identical to WT in regards to voltage-dependence and capsaicin sensitivity.